# Frequency-enhanced Data Augmentation for Vision-and-Language Navigation

**Keji He**[1,2,3]   **Chenyang Si**[*4]   **Zhihe Lu**[3]   **Yan Huang**[1,2]   **Liang Wang**[*1,2]   **Xinchao Wang**[*3]

[1]Center for Research on Intelligent Perception and Computing
National Key Laboratory for Multi-modal Artificial Intelligence Systems
Institute of Automation, Chinese Academy of Sciences
[2]School of Artificial Intelligence, University of Chinese Academy of Sciences
[3]National University of Singapore
[4]Nanyang Technological University
keji.he@cripac.ia.ac.cn    chenyang.si@ntu.edu.sg

{zhihelu, xinchao}@nus.edu.sg    {yhuang, wangliang}@nlpr.ia.ac.cn

## Abstract

Vision-and-Language Navigation (VLN) is a challenging task that requires an agent to navigate through complex environments based on natural language instructions. In contrast to conventional approaches, which primarily focus on the spatial domain exploration, we propose a paradigm shift toward the Fourier domain. This alternative perspective aims to enhance visual-textual matching, ultimately improving the agent's ability to understand and execute navigation tasks based on the given instructions. In this study, we first explore the significance of high-frequency information in VLN and provide evidence that it is instrumental in bolstering visual-textual matching processes. Building upon this insight, we further propose a sophisticated and versatile Frequency-enhanced Data Augmentation (FDA) technique to improve the VLN model's capability of capturing critical high-frequency information. Specifically, this approach requires the agent to navigate in environments where only a subset of high-frequency visual information corresponds with the provided textual instructions, ultimately fostering the agent's ability to selectively discern and capture pertinent high-frequency features according to the given instructions. Promising results on R2R, RxR, CVDN and REVERIE demonstrate that our FDA can be readily integrated with existing VLN approaches, improving performance without adding extra parameters, and keeping models simple and efficient. The code is available at https://github.com/hekj/FDA.

## 1   Introduction

Vision-and-Language Navigation (VLN) requires an embodied agent to navigate in complex environments following human instructions. This research area has garnered substantial attention due to its potential applicability in human-robot interaction contexts, such as service and rescue robotics. Despite the significant advancements [5, 12, 23, 25, 55, 34, 9, 42, 10, 37, 64, 50, 20, 68, 60] achieved in devising deep learning models for VLN, the task remains arduous, primarily due to the complexities involved in attaining precise visual-textual matching.

Recent studies in the field of VLN have focused on enhancing embodied agents' ability to navigate complex environments using natural language instructions. These studies aim to develop more robust and efficient deep learning models that can tackle the challenges of visual-textual matching, thereby

---

[*]Corresponding author

37th Conference on Neural Information Processing Systems (NeurIPS 2023).

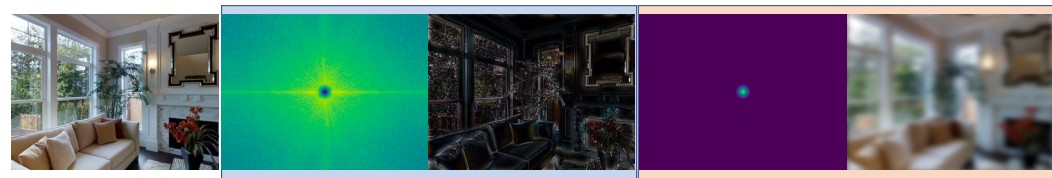

Figure 1: Examples of the high-frequency and low-frequency information. Blue background part is the high-frequency spectrum and high-frequency information in spatial domain after inverse Fourier Transform. Orange background part is about these two contents of the low-frequency information.

improving the agent's capacity to understand and execute navigation tasks based on given instructions. Some of the recent studies have investigated attention mechanisms [58, 25, 9, 64], detection models [44, 1, 10], and fine-grained trajectory-instruction pairs [66, 24, 21] to enhance cross-modal matching from the spatial domain perspective.

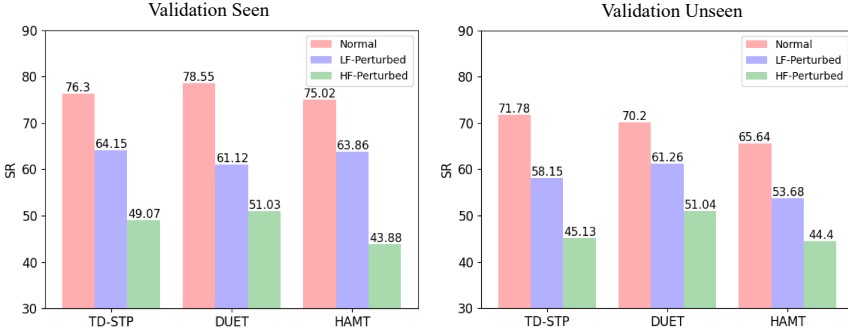

Figure 2: Sensitivity analysis of benchmark methods to high and low-frequency information, *i.e.*, HAMT [9], DUET [10], and TD-STP [64]. Normal denotes the normal navigation scenes. HF-Perturbed and LF-Perturbed denote the navigation scenes whose high-frequency and low-frequency have been perturbed, respectively.

Beyond the spatial domain perspective, in this paper, we are interested in investigating the effectiveness of other domain perspectives for VLN. We present a paradigm shift toward the Fourier domain to enhance visual textual matching, a research area that has received limited prior investigation. Specifically, high-frequency and low-frequency information pertains to distinct components of an image when analyzed within the Fourier domain. High-frequency information encompasses rapid changes, fine details, edges, and texture patterns, as illustrated in Figure 1 (part of the blue background). Conversely, low-frequency information signifies the overall structure and global features of an image, including large shapes and smooth color gradients, thereby capturing the general appearance and layouts, as shown in Figure 1 (part of the orange background).

We simply investigate the sensitivity of benchmark methods to low and high-frequency information by perturbing the low-frequency or high-frequency components in images. Three powerful baseline models, *i.e.*, HAMT [9], DUET [10], and TD-STP [64], are used to analyze the significance of low/high-frequency information on both R2R validation seen and unseen splits, wherein the navigation views are disrupted in the Fourier domain. As Shown in Figure 2, the three models maintain a relatively high Success Rate (SR) under low-frequency perturbations. However, the SR on validation seen and unseen splits decrease markedly under the high-frequency perturbation navigation process, exhibiting lower SR metrics compared to low-frequency perturbations. For example, on the validation seen dataset, the SR results for HAMT [9], DUET [10], and TD-STP [64] decrease by 31.44, 27.52, and 27.23, respectively, when affected by high-frequency perturbations. These findings reveal that VLN approaches exhibit a marked sensitivity to high-frequency information.

In light of these observations, we hypothesize that high-frequency information from images may be of paramount importance for cross-modal navigation tasks. Through extensive experiments, we ascertain that high-frequency information indeed serves a pivotal role in enriching visual representations, thereby contributing to a marked enhancement in the navigation performance of models when

faced with novel environments. To efficiently leverage the benefits of high-frequency information, we further propose a Frequency-enhanced Data Augmentation (FDA) tailored for VLN, a simplistic yet efficacious approach to augment the model's ability to capture essential high-frequency information. Specifically, the FDA method utilizes the Discrete Fourier Transform on navigation views, extracting high and low-frequency components from RGB channels. It replaces part of the high-frequency components with those from an interference image to introduce high-frequency perturbations. Augmented data is obtained by applying inverse Fourier Transform to the combination of the perturbed high-frequency and original low-frequency components. By training the agent to match original instructions with both the original and augmented navigation views concurrently, the FDA method encourages the agent to hone its ability to capture the relevant high-frequency information that best aligns with the given instruction. This, in turn, enhances the agent's overall performance in VLN tasks, making it more adept at identifying and leveraging critical high-frequency features.

Leveraging FDA during training, existing SoTA methods can achieve significant performance improvements without the need for additional parameters. This demonstrates the effectiveness of FDA in enhancing visual-textual matching capabilities in Vision-and-Language Navigation (VLN) tasks while maintaining model simplicity and efficiency. Our contributions are summarized as follows: 1) we conduct a first-of-its-kind in-depth analysis of frequency domain information in VLN task, highlighting the importance of high-frequency information in improving navigation performance. This novel perspective offers fresh research opportunities for the community to explore and enhance VLN models. 2) We further introduce a simple, effective data augmentation method, Frequency-enhanced Data Augmentation (FDA), which enhances a model's ability to discern and capture essential high-frequency information without added complexity, offering practical solutions for the research community. 3) Our method has achieved superior results on various cross-modal navigation tasks, *i.e.*, R2R, RxR, CVDN and REVERIE, and shown great adaptability across different models.

## 2    Frequency Perspective for Vision-and-Language Navigation

In this section, we first introduce the VLN problem formulation. Then we explore the potential value of high-frequency and low-frequency information for VLN. Finally, we present the Frequency-enhanced Data Augmentation (FDA) for enhancing model's ability to capture essential high-frequency information.

### 2.1    Problem Formulation

According to the VLN setup, an agent navigates in an in-door environment $E = \{p^1, p^2, ..., p^{|E|}\}$ with numbers of preset points $p^i$ in it, following a human instruction $T = \{w_1, w_2, ..., w_{|T|}\}$. Assuming at step t, the agent standing on the point $p_t^i$ can perceive the surrounding panoramic vision $O_t = (o_t^k)_{k=1}^{36}$ around it which is composed by 36 discrete observations $o_t^k$. Each observation $o_t^k = (I_t^k, \theta_t^k, \phi_t^k)$ is the combination of the $k_{th}$ view $I_t^k$ with its relative heading $\theta_t^k$ and elevation $\phi_t^k$ to the oriented view. The neighboring navigable points $N(p_t^i)$ distribute among some of these views. The agent selects the next point from these neighboring points $N(p_t^i)$ based on the correlation between instruction $T$ and the observations $o_t^k$ where $N(p_t^i)$ locate in. Then, the agent will be teleported to that selected point. This navigation continues until the agent predicts a stop action or exceeds a preset step threshold. Navigation is considered a success when the agent stops within 3 meters of the target destination.

### 2.2    High Frequency or Low Frequency: Which Benefits VLN Performance?

Considering the observations of Figure 2, we posit that high-frequency information in images may be of paramount importance for cross-modal navigation tasks. In order to validate this hypothesis, we conduct a straightforward experiment that entails the fusion of original image features with their corresponding high-frequency or low-frequency components. These amalgamated features subsequently serve as input for the navigation network during training and testing, as shown in Figure 3. The results on TD-STP [64] are present in Table 1. We can observe the integration of high-frequency features leads to an improvement in navigation accuracy, such as SR increasing from 69.65 to 70.37 for unseen environments. Conversely, incorporating low-frequency features appears to have an adverse effect, resulting in a decrease in accuracy with SR dropping from 69.65 to 68.75. The above observation highlights the critical role of high-frequency information in cross-modal navigation. This is because 1) high-frequency information encompasses fine details such as edges, corners, and texture patterns. These details are essential for accurately identifying and differentiating objects, scenes, and locations, which can lead to more effective visual-textual matching and better

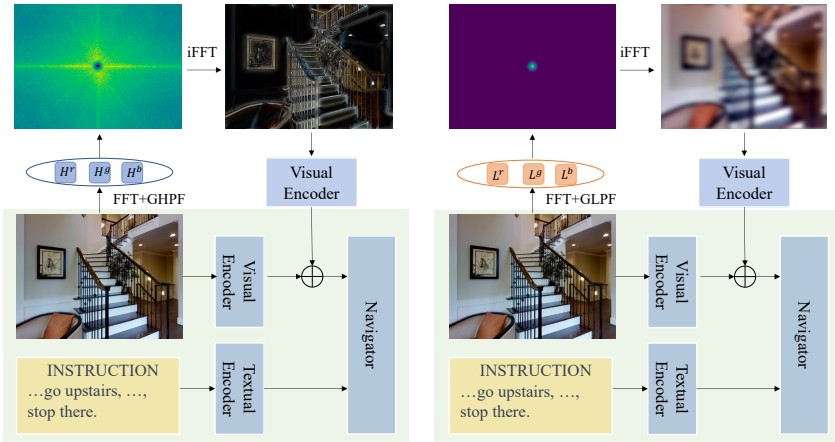

(a) Navigator with high-frequency information as extra input      (b) Navigator with low-frequency information as extra input

Figure 3: Integrate high-frequency and low-frequency information into the navigator (TD-STP). FFT and iFFT denote the Fourier Transform and inverse Fourier Transform. GHPF and GLPF denote the Gaussian High-Pass Filter and Gaussian Low-Pass Filter, respectively. The green background part is the pipeline of the baseline model (TD-STP).

Table 1: Results on R2R task: integrating high-frequency (HF) and low-frequency (LF) information.

| | Validation Seen | | | | Validation Unseen | | | |
|---|---|---|---|---|---|---|---|---|
| | TL | NE↓ | SR↑ | SPL↑ | TL | NE↓ | SR↑ | SPL↑ |
| TD-STP | 12.66 | 2.53 | 76.30 | 71.70 | 13.53 | 3.28 | 69.65 | 62.97 |
| TD-STP+LF | 12.83 | 2.42 | 76.49 | 71.36 | 14.29 | 3.30 | 68.75 | 61.96 |
| TD-STP+HF | 12.29 | **2.30** | **78.75** | **73.78** | 13.68 | **3.14** | **70.37** | **63.62** |

navigation performance. 2) Models trained with high-frequency information tend to be more robust to environmental variations and exhibit greater generalization ability to unseen environments, as the model learns to focus on a more diverse set of features, rather than just memorizing specific low-frequency, global patterns present in the training data.

## 2.3 Frequency-enhanced Data Augmentation

To systematically capitalize on the advantages of high-frequency information, we subsequently introduce a Frequency-enhanced Data Augmentation (FDA) method, specifically designed for VLN tasks. This approach is both straightforward and effective in bolstering the model's capacity to apprehend and incorporate crucial high-frequency information. As illustrated in Figure 4, the reference image $I$ is a navigation view corresponding to the navigation instruction $T$ ("*Go forward, pass by the refrigerator, turn left behind the dining table, go straight through the doorway...*"). The interference image $\hat{I}$ is another navigation view sampled from the Matterport3d (Mp3d) dataset [6] randomly. To prevent information leakage, all interference images are sampled from training/validation seen splits, and no image from the validation unseen and test splits is used. First, we transform these two images to the frequency domain space via FFT, resulting in two frequency spectrums $F_I^{\{rgb\}}$ and $F_{\hat{I}}^{\{rgb\}}$:

$$F_I^{\{rgb\}} = \mathcal{F}^{\{rgb\}}(I), \ \ F_{\hat{I}}^{\{rgb\}} = \mathcal{F}^{\{rgb\}}(\hat{I}) \tag{1}$$

where $\mathcal{F}^{\{rgb\}}$ denotes the Fourier Transform on RGB color channels. Afterward, we apply High-Pass and Low-Pass Gaussian Filters on the two frequency spectrums to obtain the reference high-frequency $H^{\{rgb\}}$, reference low-frequency $L^{\{rgb\}}$ and interference high-frequency $\hat{H}^{\{rgb\}}$.

$$H^{\{rgb\}} = \mathcal{G}_h \odot F_I^{\{rgb\}}, \ \ L^{\{rgb\}} = \mathcal{G}_l \odot F_I^{\{rgb\}}, \ \ \hat{H}^{\{rgb\}} = \mathcal{G}_h \odot F_{\hat{I}}^{\{rgb\}} \tag{2}$$

where $\mathcal{G}_h$ and $\mathcal{G}_l$ denote the Gaussian High-Pass Filter (GHPF) and Gaussian Low-Pass Filter (GLPF), and $\odot$ is element-wise multiplication. Then, we mix the high-frequency components of these two

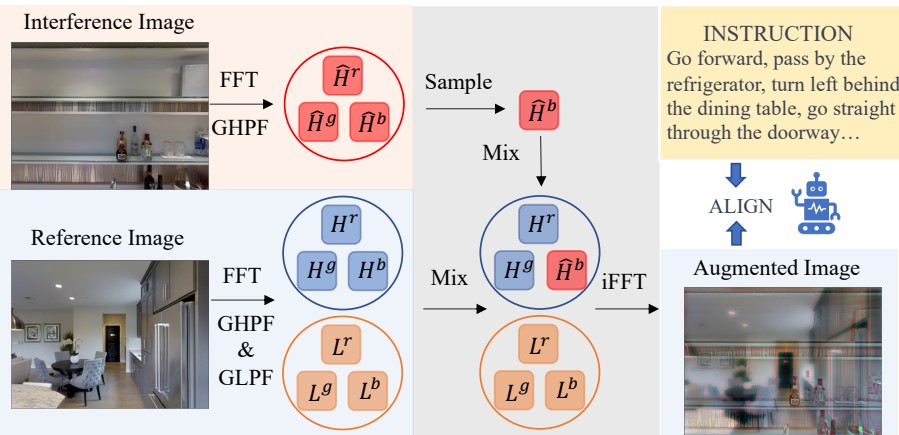

Figure 4: Our approach mixes the interference high-frequency and the reference high-frequency together. Then the augmented image is obtained by applying inverse Fourier Transform on the combination of the mixed high-frequency and reference low-frequency. Finally, the augmented image containing only a portion of the reference high-frequency is aligned to the instruction to encourage agent to discern and capture the reference high-frequency information. FFT and iFFT represent the Fourier Transform and inverse Fourier Transform. GHPF and GLPF denote the Gaussian High-Pass Filter and Gaussian Low-Pass Filter, respectively.

images. Specifically, for each RGB channel of the reference image, there is a certain probability that its high-frequency component is replaced by the interference high-frequency from the same channel:

$$H_{mix}^c = \mathcal{M}ix(H^c, \hat{H}^c) = \begin{cases} H^c, & probability \ of \ 1/3 \\ \hat{H}^c, & others \end{cases}, \ c \in \{r, g, b\} \tag{3}$$

$$H_{mix}^{\{rgb\}} = \mathcal{M}ix(H^{\{rgb\}}, \hat{H}^{\{rgb\}}) \tag{4}$$

where $H_{mix}^{\{rgb\}}$ is the mixed high frequency. We combine it with the reference low frequency $L^{\{rgb\}}$ and then apply iFFT to obtain the frequency-enhanced augmented image $I_{mix}$:

$$I_{mix} = \mathcal{F}^{-1}(F_{mix}^{\{rgb\}}) = \mathcal{F}^{-1}(H_{mix}^{\{rgb\}}, L^{\{rgb\}}). \tag{5}$$

Finally, the original image $I$ and augmented image $I_{mix}$ share the same textual instruction label $T$ and are used alternately to train the agent during the training phase:

$$L(\theta) = \begin{cases} NavigatorLoss(I, T, \theta), & odd\text{-}numbered \ step \\ NavigatorLoss(I_{mix}, T, \theta), & even\text{-}numbered \ step \end{cases} \tag{6}$$

where $L(\theta)$ denotes the navigation loss considering both the original image $I$ and frequency-enhanced augmented image $I_{mix}$, $\theta$ denotes the parameters of the navigator.

## 3 Experiments

### 3.1 Datasets

The visual environments are based on the photo-realistic dataset Matterport3d (Mp3d) [6]. There are a total of 90 houses, with 61, 11, and 18 houses allocated for training/validation seen, validation unseen, and test splits, respectively. Four datasets containing the instruction-trajectory pairs have been adopted: R2R [5], RxR [29], CVDN [52] and REVERIE [45]. R2R includes 7,189 trajectories and each trajectory has 3 corresponding English instructions. RxR has 16,522 trajectories and 126,069 multilingual (English, Hindi and Telugu) instructions in total. Compared to R2R, its trajectories are longer and instructions are more complex. CVDN is based on human-to-human dialogs. The multi-turn dialogs instruct the agent to navigate to the destinations. There are 2,050 dialogs and over 7,000 trajectories in this dataset. REVERIE contains a total of 21,702 high-level instructions, including descriptions of navigation destinations and the objects to be found.

## 3.2 Evaluation Metrics

According to the conventions of each task, we adopt a total of ten evaluation metrics, namely Success Rate (SR), Success Rate weighted by Path Length (SPL), Trajectory Length (TL), Navigation Error (NE), normalized Dynamic Time Warping (nDTW), Success weighted by normalized Dynamic Time Warping (SDTW), Goal Progress (GP), Navigation Oracle Success Rate (OSR), Remote Grounding Success Rate (RGS) and Remote Grounding Success Rate weighted by Navigation Path Length (RGSPL). SR measures the proportion of successful navigation. SPL is a trade-off metric between SR and TL. TL is the average trajectory length. NE measures the average distance between the final position and the destination. nDTW measures the fidelity between the predicted trajectory and the groundtruth trajectory. SDTW combines the evaluation of SR with nDTW. GP measures the progress toward the goal. OSR measures the proportion of successful arrivals at or passage through the destinations. RGS represents the proportion of successful groundings to the correct object. RGSPL is a trade-off metric between RGS and TL. R2R uses the SR, SPL, TL and NE metrics. RxR adopts the SR, SPL, nDTW and SDTW metrics. CVDN uses the GP metric. REVERIE adopts the TL, OSR, SR, SPL, RGS and RGSPL metrics.

## 3.3 Ablation Study

Table 2: Models enhanced by the FDA method on R2R task.

|  |  | Validation Unseen | | | |
| --- | --- | --- | --- | --- | --- |
|  |  | TL | NE↓ | SR↑ | SPL↑ |
| Transformer-based Models | HAMT | 11.50 | 3.57 | 65.64 | 60.15 |
|  | HAMT+FDA | 11.86 | **3.44** | **67.52** | **62.16** |
|  | DUET | 12.95 | 3.38 | 70.20 | 60.28 |
|  | DUET+FDA | 13.61 | **3.16** | **71.86** | **61.16** |
|  | TD-STP | 13.53 | 3.28 | 69.65 | 62.97 |
|  | TD-STP+FDA | 13.68 | **3.02** | **71.78** | **64.00** |
| LSTM-based Models | Follower | - | - | 38.3 | - |
|  | Follower+FDA | - | - | **45.6** | - |
|  | EnvDrop | 10.08 | 5.15 | 51.1 | 47.8 |
|  | EnvDrop+FDA | 13.43 | **5.00** | **54.5** | **49.1** |

**FDA Assists in the R2R Task.** We enhance both Transformer-based and LSTM-based strong baselines, *i.e.*, HAMT [9], DUET [10], TD-STP [64], Follower [15] and EnvDrop [49], with our FDA method on R2R task, as shown in Table 2. The three enhanced Transformer-based models have achieved clear advantages over the baselines, with SR improved by 1.88, 1.66, 2.13 and SPL improved by 2.01, 0.88, 1.02, respectively on the validation unseen split. Meanwhile, our FDA method can also boost the performances of the two LSTM-based models, with SR improved by 7.3 and 3.4, respectively. These results demonstrate the adaptability of our approach across various models, which can significantly improve their performances and alleviate overfitting.

Table 3: Ablations on the RxR and CVDN tasks.

|  | RxR | | | | CVDN |
| --- | --- | --- | --- | --- | --- |
|  | nDTW↑ | SDTW↑ | SR↑ | SPL↑ | GP↑ |
| HAMT | 64.3 | 49.6 | 58.1 | 54.0 | 5.03 |
| HAMT+FDA | **66.4** | **51.6** | **59.8** | **55.8** | **5.39** |

**FDA Facilitates Performances on Other Cross-Modal Navigation Tasks.** Furthermore, we evaluate our FDA method on RxR, CVDN and REVERIE tasks. Table 3 shows the results on validation unseen splits of RxR and CVDN tasks. The HAMT equipped with FDA has obtained notable superior performances compared to the baseline. On RxR task, the SR and SDTW have been improved by 1.7 and 2.0, respectively. On CVDN task, our method could boost the GP performance by 0.36. Table 4 shows our method's performance on REVERIE task. The DUET enhanced by FDA significantly outperforms the baseline, with SR and RGS improved by 2.92 and 3.49. These observations indicate

Table 4: Ablabtion on the REVERIE task.

| | Validation Unseen | | | | | |
|---|---|---|---|---|---|---|
| | TL | OSR↑ | SR↑ | SPL↑ | RGS↑ | RGSPL↑ |
| DUET | 18.76 | 48.59 | 44.65 | 32.92 | 28.57 | 21.01 |
| DUET+FDA | 19.04 | **51.41** | **47.57** | **35.90** | **32.06** | **24.31** |

Table 5: Performances in navigation scenes with high-frequency interference on R2R task. Red background part represents the baselines in normal navigation scenes. White and gray background parts represent the baselines and our augmented models in scenes with high-frequency interference. Note that this evaluation setting is non-standard R2R, and is specially designed to validate our method.

| | Validation Seen | | | | Validation Unseen | | | |
|---|---|---|---|---|---|---|---|---|
| | TL | NE↓ | SR↑ | SPL↑ | TL | NE↓ | SR↑ | SPL↑ |
| HAMT | 11.06 | 2.49 | 75.02 | 71.78 | 11.50 | 3.57 | 65.64 | 60.15 |
| [MP3d] HAMT | 15.47 | 3.61 | 43.88 | 38.94 | 15.76 | 5.55 | 44.40 | 36.93 |
| [MP3d] HAMT+FDA | 11.99 | **3.40** | **67.38** | **62.93** | 13.67 | **4.40** | **58.32** | **51.12** |
| [ImageNet] HAMT | 17.19 | 6.05 | 41.23 | 33.37 | 18.60 | 6.10 | 38.87 | 31.04 |
| [ImageNet] HAMT+FDA | 13.26 | **4.24** | **59.06** | **52.90** | 13.43 | **4.67** | **55.56** | **48.99** |
| DUET | 11.73 | 2.38 | 78.55 | 73.71 | 12.95 | 3.38 | 70.20 | 60.28 |
| [MP3d] DUET | 16.96 | 5.31 | 51.03 | 41.30 | 15.75 | 5.19 | 51.04 | 39.20 |
| [MP3d] DUET+FDA | 14.06 | **3.17** | **69.44** | **60.78** | 14.69 | **3.74** | **65.73** | **53.16** |
| [ImageNet] DUET | 17.39 | 6.11 | 43.29 | 33.41 | 16.79 | 5.78 | 44.61 | 32.72 |
| [ImageNet] DUET+FDA | 17.01 | **3.90** | **62.49** | **59.79** | 16.89 | **3.94** | **63.69** | **49.78** |
| TD-STP | 12.66 | 2.53 | 76.30 | 71.70 | 13.53 | 3.28 | 69.65 | 62.97 |
| [MP3d] TD-STP | 21.34 | 5.23 | 49.07 | 40.62 | 21.25 | 5.73 | 45.13 | 37.41 |
| [MP3d] TD-STP+FDA | 13.50 | **3.04** | **69.93** | **63.44** | 16.65 | **4.02** | **62.11** | **52.77** |
| [ImageNet] TD-STP | 21.81 | 5.91 | 40.55 | 32.95 | 21.47 | 5.74 | 42.10 | 34.17 |
| [ImageNet] TD-STP+FDA | 16.20 | **3.69** | **63.08** | **54.74** | 16.80 | **4.02** | **61.13** | **51.37** |

that our approach can be easily and effectively integrated into other cross-modal navigation tasks while ensuring a significant performance boost.

**Performances on Navigation Scenes with High-Frequency Interference.** As analyzed before, existing VLN methods are severely limited in high-frequency perturbed navigation scenes. Table 5 shows more details. We observe that the baselines can perform well under the normal navigation scenes. **Setting 1:** Then, we introduce high-frequency interference, whose corresponding spatial image is the navigation views randomly sampled from MP3d [6], into seen/unseen navigation views as described in the method section to get the high-frequency perturbed unseen navigation scenes. Navigating under such scenes, all models have experienced a significant decrease in SR metric. Taking TD-STP as an example, its SR decreases by as much as 27.23 and 24.52 on validation seen and validation unseen, respectively. The fact that the models are easily and severely affected by high-frequency interference suggests their lack of ability to effectively identify and capture the necessary high-frequency information based on the instructions. However, with the assistance of our FDA, all models show notable improvement in performance. HAMT, DUET, and TD-STP see SR metric increases of 23.5, 18.41, 19.96 on validation seen split, and 13.92, 14.69, 16.98 on validation unseen split. **Setting 2:** To distinguish the source of interference from the training phase, we have also sampled high-frequency interference from the 1000 classes of ImageNet [48]. Due to the out-of-domain interference, we find that the SR of baseline models is significantly more affected under this setting. Taking TD-STP as an example, its SR on validation seen and validation unseen decreases by 35.75 and 27.55, respectively. In such challenging scenes, our FDA-enhanced models whose training data only contains high-frequency and low-frequency information from Mp3d, perform remarkably well, improving the SR of HAMT, DUET, and TD-STP by 17.83, 19.2, and 22.53 on validation seen, and by 16.69, 19.08, and 19.03 on validation unseen, respectively. In most cases, the navigation performances of the models have gained even greater improvements compared to the first setting. These two settings illustrate FDA-enhanced models excelling in both the seen/unseen perturbed images (val seen/unseen) and the seen/unseen perturbation images (MP3d/ImageNet). This serves as compelling evidence of our method's remarkable capability to recognize and capture essential high-frequency information to boost navigation performance.

Table 6: Comparison with the visual and textual data augmentation methods on R2R task.

| | | Validation Unseen | | | |
|---|---|---|---|---|---|
| | | TL | NE↓ | SR↑ | SPL↑ |
| Visual Data Augmentation | HAMT | 11.50 | 3.57 | 65.64 | 60.15 |
| | HAMT+ENVEDIT | 11.15 | 3.48 | 67.31 | **62.84** |
| | HAMT+FDA | 11.86 | **3.44** | **67.52** | 62.16 |
| | TD-STP | 13.53 | 3.28 | 69.65 | 62.97 |
| | TD-STP+ENVEDIT | 14.70 | 3.20 | 69.82 | 62.55 |
| | TD-STP+FDA | 13.68 | **3.02** | **71.78** | **64.00** |
| Textual Data Augmentation | Follower | - | - | 38.3 | - |
| | Follower+FDA | - | - | **45.6** | - |
| | Follower+Speaker | - | - | 40.4 | - |
| | Follower+Speaker+FDA | - | - | **46.3** | - |
| | TD-STP$^-$ | 13.78 | 3.62 | 65.73 | 58.72 |
| | TD-STP$^-$+FDA | 14.77 | **3.44** | **66.96** | **59.72** |
| | TD-STP$^-$+PREVALENT | 13.53 | 3.28 | 69.65 | 62.97 |
| | TD-STP$^-$+PREVALENT+FDA | 13.68 | **3.02** | **71.78** | **64.00** |

**Comparison with Existing Data Augmentation Methods.** Table 6 shows the comparison with the visual data augmentation method ENVEDIT[2] [33], and textual data augmentation methods Speaker [15] and PREVALENT [20]. Compared to ENVEDIT which aims to increase the diversity of the environment by using GANs to alter environment style, our method, which also serves as a visual data augmentation method, demonstrates a clear superiority, with SR of HAMT/TD-STP increased by 0.21/1.96 on validation unseen, respectively. In comparison with the textual augmentation methods, the results indicate that our FDA method demonstrates consistent improvements on both baseline models Follower and TD-STP$^-$[3]. Additionally, it effectively complements the two textual augmentation methods, further enhancing cross-modal navigation performance. Our FDA has demonstrated superior performance without relying on external models such as visual/textual generative models. As a result, our method could greatly reduce the complexity of data augmentation work in VLN field.

### 3.4 Comparison with the State-of-the-art Methods

We compare our method with SoTA methods on R2R, RxR, CVDN and REVERIE tasks.

On **R2R** task, Table 7 shows our method achieves the best performance on both validation unseen and test splits. Compared to the DUET model which has been fed with the extra object features not included in our method, the SR/SPL of our method still outperforms it by 2/4 points on validation unseen. Compared to TD-STP, Our method shows SR/SPL improvements of 2/1 and 3/2 points on validation unseen and test splits. On **RxR** task, Table 8 shows the comparisons between our method and previous methods. The results show that our method outperforms the previous SoTA HAMT with SPL/nDTW improved by 1.8/2.1 and 0.4/1.5 on validation unseen and test splits, respectively. On **CVDN** task, our method also shows superior performance over previous methods in Table 9. Compared to HAMT, the previous SoTA, our method gains a GP improvement of 0.05/0.36/0.25 on validation seen, validation unseen and test splits, respectively. On **REVERIE** task, as shown in Table 10, our method outperforms the previous SoTA DUET with the increase of 2.92/2.98/3.49/3.30 and 1.21/0.49/2.03/1.47 in SR/SPL/RGS/RGSPL on validation unseen and test splits, respectively.

The remarkable performance analyzed above has demonstrated the excellence of our method in enhancing cross-modal navigation, as well as the outstanding adaptability across different tasks.

## 4 Related Work

**Vision-and-Language Navigation.** Anderson *et al.* [5] are the first to introduce the Vision-and-Language Navigation, where an agent is asked to follow an English instruction in R2R to navigate.

---

[2]Unlike ENVEDIT, our method operates without external models. To ensure fairness, we compare our method with ENVEDIT's Editing Style, which relies on the fewest external models.

[3]TD-STP$^-$ denotes the TD-STP model after excluding data augmentation PREVALENT.

Table 7: Comparison with SoTA methods on R2R task.

| | Validation Unseen | | | | Test Unseen | | | |
|---|---|---|---|---|---|---|---|---|
| | TL | NE↓ | SR↑ | SPL↑ | TL | NE↓ | SR↑ | SPL↑ |
| Seq2Seq [5] | 8.39 | 7.81 | 22 | - | 8.13 | 7.85 | 20 | 18 |
| Speaker-Follower [15] | 6.62 | - | 35 | - | 14.82 | 6.62 | 35 | 28 |
| SMNA [39] | - | 5.52 | 45 | 32 | 18.04 | 5.67 | 48 | 35 |
| RCM [58] | 11.46 | 6.09 | 43 | - | 11.97 | 6.12 | 43 | 38 |
| EGP [12] | - | 4.83 | 56 | 44 | - | 5.34 | 53 | 42 |
| EnvDrop [51] | 10.70 | 5.22 | 52 | 48 | 11.66 | 5.23 | 51 | 47 |
| PREVALENT [20] | 10.19 | 4.71 | 58 | 53 | 10.51 | 5.30 | 54 | 51 |
| RelGraph [23] | 9.99 | 4.73 | 57 | 53 | 10.29 | 4.75 | 55 | 52 |
| RecBert [25] | 12.01 | 3.93 | 63 | 57 | 12.35 | 4.09 | 63 | 57 |
| Airbert [19] | 11.78 | 4.01 | 62 | 56 | 12.41 | 4.13 | 62 | 57 |
| DUET [10] | 13.94 | 3.31 | 72 | 60 | 14.73 | 3.65 | 69 | 59 |
| DUET* [10] | 12.95 | 3.38 | 70 | 60 | - | - | - | - |
| HAMT [9] | 11.46 | 2.29 | 66 | 61 | 12.27 | 3.93 | 65 | 60 |
| HAMT* [9] | 11.50 | 3.57 | 66 | 60 | - | - | - | - |
| TD-STP [64] | - | 3.22 | 70 | 63 | - | 3.73 | 67 | 61 |
| TD-STP* [64] | 13.53 | 3.28 | 70 | 63 | 14.71 | 3.79 | 66 | 60 |
| Ours (TD-STP + FDA) | 13.68 | **3.02** | **72** | **64** | 14.76 | **3.41** | **69** | **62** |

Table 8: Comparison with SoTA methods on RxR task.

| | Validation Unseen | | | | Test Unseen | | | |
|---|---|---|---|---|---|---|---|---|
| | nDTW↑ | SDTW↑ | SR↑ | SPL↑ | nDTW↑ | SDTW↑ | SR↑ | SPL↑ |
| Multilingual Baseline [29] | 38.9 | 18.2 | 22.8 | - | 36.8 | 16.9 | 21.0 | 18.6 |
| Monolingual Baseline [29] | 44.5 | 23.1 | 28.5 | - | 41.1 | 20.6 | 25.4 | 22.6 |
| SAA [34] | 52.7 | 33.0 | 40.0 | 36.0 | 46.8 | 29.1 | 35.4 | 31.6 |
| EnvDrop + CLIP-ViL [49] | 55.7 | - | 42.6 | - | 51.1 | 32.4 | 38.3 | 35.2 |
| HAMT [9] | 63.1 | 48.3 | 56.5 | 56.0 | 59.9 | 45.2 | 53.1 | 46.6 |
| HAMT* [9] | 64.3 | 49.6 | 58.1 | 54.0 | 60.5 | 47.0 | **55.5** | 48.4 |
| Ours (HAMT+FDA) | **66.4** | **51.6** | **59.8** | **55.8** | **62.0** | **47.3** | **55.5** | **48.8** |

After that, model architecture [58, 4, 24, 12, 23, 25, 55, 1, 34, 9, 42, 10, 37, 7, 59, 2, 3, 22], strategy design [28, 40, 39, 17, 8, 18], training paradigm [66, 62, 54, 38], loss design [65, 36, 32], reward shaping [27, 56, 21], pretraining [35, 20, 41, 46, 49, 61, 47] and probing experiments [26, 63, 67] have been widely explored. Then, a more challenging task RxR [29] containing multilingual instructions and longer trajectories is proposed. A series of works [34, 32, 21, 9] have explored the above and make significant progress. CVDN [52] is a task asking an agent to navigate following multi-turn human-to-human dialog. In addition to requiring the agent to have cross-modal understanding, it also demands a certain level of reasoning ability. Works [50, 20, 69, 68, 60, 9] focusing on that have been investigated to advance the development of this field. While these works have conducted extensive exploration, they have almost neglected the frequency domain that our work has demonstrated to be crucial for VLN navigation.

Table 9: Comparison with SoTA methods on CVDN task.

| | Val Seen | Val Unseen | Test Unseen |
|---|---|---|---|
| PREVALENT [20] | - | 3.15 | 2.44 |
| CMN [69] | 7.05 | 2.97 | 2.95 |
| VISITRON [50] | 5.11 | 3.25 | 3.11 |
| SCoA [68] | 7.11 | 2.85 | 3.31 |
| MT-RCM+EnvAg [60] | 5.07 | 4.65 | 3.91 |
| HAMT [9] | 6.91 | 5.13 | 5.58 |
| HAMT* [9] | 7.43 | 5.03 | 5.43 |
| Ours (HAMT+FDA) | **7.48** | **5.39** | **5.68** |

Table 10: Comparison with SoTA methods on REVERIE task.

| | Validation Unseen | | | | | | Test Unseen | | | | | |
|---|---|---|---|---|---|---|---|---|---|---|---|---|
| | TL | OSR↑ | SR↑ | SPL↑ | RGS↑ | RGSPL↑ | TL | OSR↑ | SR↑ | SPL↑ | RGS↑ | RGSPL↑ |
| RecBERT [25] | 16.78 | 35.02 | 30.67 | 24.90 | 18.77 | 15.27 | 15.86 | 32.91 | 29.61 | 23.99 | 16.50 | 13.51 |
| Airbert [19] | 18.71 | 34.51 | 27.89 | 21.88 | 18.23 | 14.18 | 17.91 | 34.20 | 30.28 | 23.61 | 16.83 | 13.28 |
| HAMT [9] | 14.08 | 36.84 | 32.95 | 30.20 | 18.92 | 17.28 | 13.62 | 33.41 | 30.40 | 26.67 | 14.88 | 13.08 |
| DUET [10] | 22.11 | 51.07 | 46.98 | 33.73 | 32.15 | 23.03 | 21.30 | 56.91 | 52.51 | 36.06 | 31.88 | 22.06 |
| DUET* [10] | 18.76 | 48.59 | 44.65 | 32.92 | 28.57 | 21.01 | 16.31 | 52.77 | 48.41 | 35.96 | 28.31 | 20.61 |
| DUET+FDA | 19.04 | **51.41** | **47.57** | **35.90** | **32.06** | **24.31** | 17.30 | **53.54** | **49.62** | **36.45** | **30.34** | **22.08** |

**Data Augmentation in Vision-and-Language Navigation.** In terms of text augmentation, Fried *et al.* [15], Wang *et al.* [57] and Dou *et al.* [14] use instruction generators to generate pseudo-instructions according to sampled trajectories. In terms of vision augmentation, Fu *et al.* [16] learn a sampler to sample the challenging paths to enhance the agent's ability for handling them. Tan *et al.* [51] use environment dropout on seen environments to mimic unseen environments and generate the pseudo-instructions for it with instruction generator. Parvaneh *et al.* [43] learn a model to generate counterfactuals on seen environments to simulate unseen environments. Li *et al.* [33] adopt GAN to transfer the environment style for increasing the diversity of the environment. However, all of these works focus on the augmentation in the spatial domain and have ignored the high-frequency information which is crucial for cross-modal matching. Moreover, they have to rely on extra generator models, which require intricate design and will increase the model burden.

**Data Augmentation in Robotics.** Domain randomization is a form of Data Augmentation (DA) that has shown effectiveness in transferring reinforcement learning policies from simulation to the real world [53]. In contrast, Cobbe *et al.* [11] propose a modification to Cutout [13], where multiple rectangular regions of varying sizes are masked with a random color for each observation. Lee *et al.* [31] use a randomized (convolutional) neural network that adds random perturbations to the input observations, which helps the trained agents learn more robust features that are invariant across diverse and randomized environments. RAD [30] investigates the general DAs for robotics on both pixel-based and state-based inputs. In this paper, we propose a simple and effective frequency-enhanced DA for the cross-modal task VLN, a fundamental task in robotics.

## 5 Conclusion

Shifting from the perspective of spatial domain previous works mainly focus on, we provide a first-of-its-kind in-depth analysis of the frequency domain information in VLN tasks. Our findings have demonstrated the importance of high-frequency information in enhancing navigation performance. In light of the findings, we further propose the Frequency-enhanced Data Augmentation (FDA) method, a simple and effective data augmentation technique to boost a model's capability to discern and capture essential high-frequency information without the need for additional auxiliary generative models. Our method on a range of VLN tasks, including R2R, RxR, CVDN and REVERIE, achieves notable performances and shows great adaptability across different VLN models. We hope our work can offer a practical solution and bring new insights to the VLN research community.

**Limitations and Future Work.** Our work focuses on enhancing the model's general ability to recognize and capture essential high-frequency information. However, we have not yet explored the fine-grained correlation between frequency and specific scenes or categories. This area of investigation remains an avenue for future exploration.

## 6 Acknowledgements

This project is supported by National Key R&D Program of China (2022ZD0117900), National Natural Science Foundation of China (62236010 and 62276261), Key Research Program of Frontier Sciences CAS Grant No. ZDBS-LYJSC032, the National Research Foundation, Singapore, under its Medium Sized Center for Advanced Robotics Technology Innovation, and the Ministry of Education, Singapore, under its Academic Research Fund Tier 2 (Award Number: MOE-T2EP20122-0006).

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
