# Frequency-enhanced Data Augmentation for Vision-and-Language Navigation
# ——— Supplemental Material ———

Keji He[1,2,3]   Chenyang Si[*4]   Zhihe Lu[3]   Yan Huang[1,2]   Liang Wang[*1,2]   Xinchao Wang[*3]

[1]Center for Research on Intelligent Perception and Computing
National Key Laboratory for Multi-modal Artificial Intelligence Systems
Institute of Automation, Chinese Academy of Sciences
[2]School of Artificial Intelligence, University of Chinese Academy of Sciences
[3]National University of Singapore
[4]Nanyang Technological University
keji.he@cripac.ia.ac.cn    chenyang.si@ntu.edu.sg
{zhihelu, xinchao}@nus.edu.sg    {yhuang, wangliang}@nlpr.ia.ac.cn

## Appendix

## A   Analysis on the Impact of Random Seeds

Table 1: Performance of different seeds on R2R task.

|        | Validation Unseen | | | |
|--------|-------|------|------|-------|
|        | TL    | NE↓  | SR↑  | SPL↑  |
| seed-1 | 13.68 | 3.02 | 71.78 | 64.00 |
| seed-2 | 14.18 | 3.02 | 71.39 | 63.63 |
| seed-3 | 14.33 | 3.03 | 72.54 | 64.04 |
| seed-4 | 15.21 | 3.01 | 71.39 | 62.88 |
| Average | 14.35 | 3.02 | 71.78 | 63.64 |

As described in the method section of the main manuscript, the interference image $\hat{I}$ is a navigation view randomly sampled from the Matterport3d dataset. Table 1 presents the impacts of different random seeds for sampling the interference images. Across the four sets of random seeds, our augmented data consistently enables the agent to achieve an SR of 71.39 (TD-STP) or higher. The average performance[2] of the augmented agents on SR metrics is 71.78. This indicates that the superior performance of our method is not sensitive to the choice of random seeds.

Table 2: Analysis on the source of the interference frequency. $FDA_l$ and FDA denote utilizing the low-frequency and high-frequency of the interference image for data augmentation, respectively.

|              | Validation Unseen | | | |
|--------------|-------|------|------|-------|
|              | TL    | NE↓  | SR↑  | SPL↑  |
| TD-STP       | 13.53 | 3.28 | 69.65 | 62.97 |
| TD-STP+$FDA_l$ | $14.35_{\pm 0.43}$ | $3.23_{\pm 0.11}$ | $69.96_{\pm 0.57}$ | $62.25_{\pm 0.29}$ |
| TD-STP+FDA   | $14.35_{\pm 0.86}$ | $3.23_{\pm 0.01}$ | $71.78_{\pm 0.76}$ | $63.64_{\pm 0.76}$ |

---

[*]Corresponding author

[2]Experiments in the main manuscript are based on seed-1 which has an average performance.

37th Conference on Neural Information Processing Systems (NeurIPS 2023).

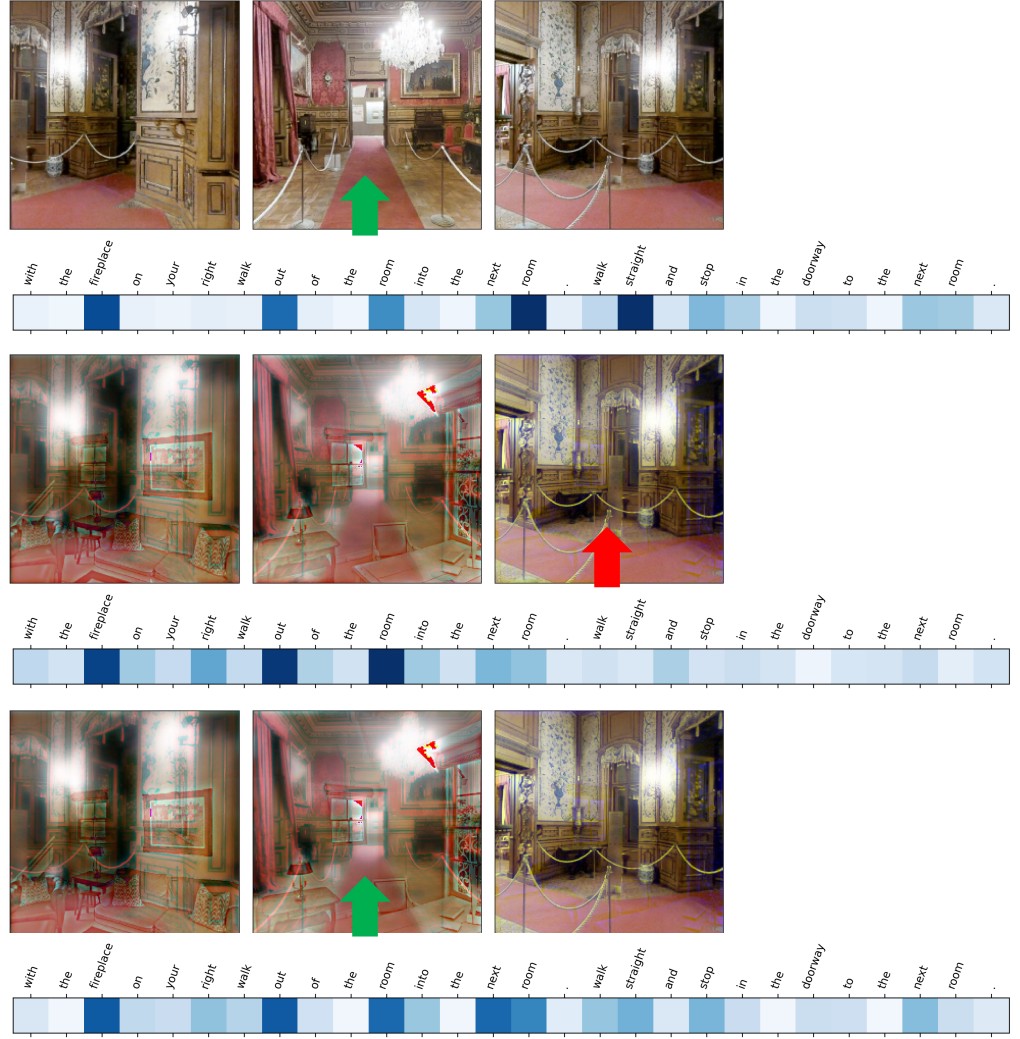

Figure 1: Navigation examples in normal and high-frequency perturbed scenes. The first row shows candidate views and attention weights of the baseline model TD-STP in normal scenes. The second row displays the same for high-frequency perturbed scenes. The third row presents the FDA-enhanced model's candidate views and attention weights in high-frequency perturbed scenes. Green and red arrows denote the correct and erroneous actions, respectively.

## B   Analysis on the Source of the Interference Frequency

Table 2 presents the effects of using high-frequency (with a cutoff frequency of 10) and low-frequency (with a cutoff frequency of 1) as interference sources in our data augmentation. We observe a significant enhancement in the model's performance when using high-frequency as the interference source, with SR/SPL increased by 2.13/0.67. In contrast, low-frequency interference yields marginal improvements, and even leads to a decrease in SPL. This result further substantiates the effectiveness of high-frequency information in enhancing cross-modal navigation, demonstrating that our approach enables the agent to better capture and utilize high-frequency information for improved navigation.

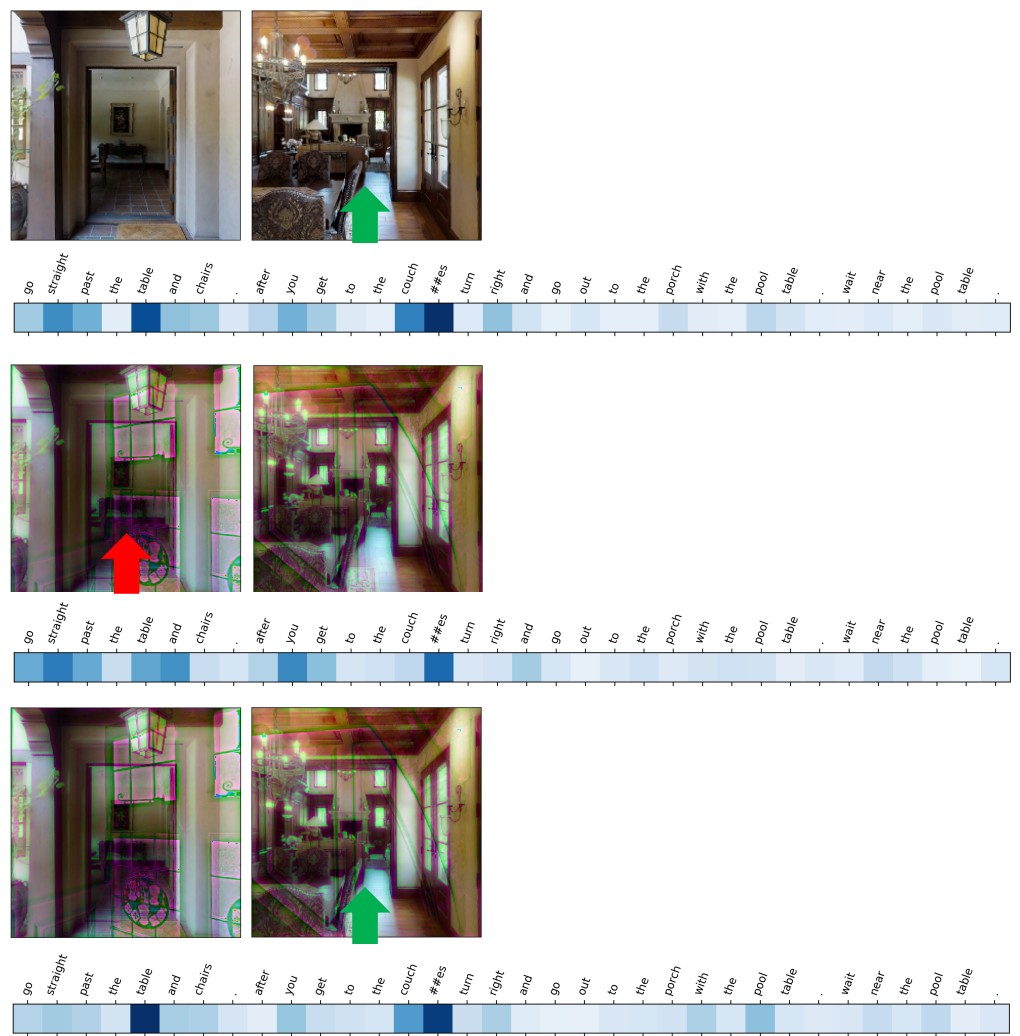

Figure 2: Navigation examples in normal and high-frequency perturbed scenes. The first row shows candidate views and attention weights of the baseline model TD-STP in normal scenes. The second row displays the same for high-frequency perturbed scenes. The third row presents the FDA-enhanced model's candidate views and attention weights in high-frequency perturbed scenes. Green and red arrows denote the correct and erroneous actions, respectively.

## C Visualization Examples

Figure 1 and Figure 2 show examples of the candidate views and textual attention heatmap in normal scenes and high-frequency perturbed scenes. It can be observed that the baseline model (the second row) often fails to pay enough attention to the correct words, such as "walk straight (Figure 1)" and "couch (Figure 2)", in scenes with high-frequency interference, leading to erroneous action selections as a result. In comparison, even in the presence of high-frequency interference, the FDA-enhanced model (the third row) can achieve textual attention similar to the baseline model navigating in normal scenes (the first row), and is capable of selecting the correct actions. This suggests that our method enables the model to effectively recognize and capture the essential high-frequency information, thereby improving text grounding and enhancing navigation performance.

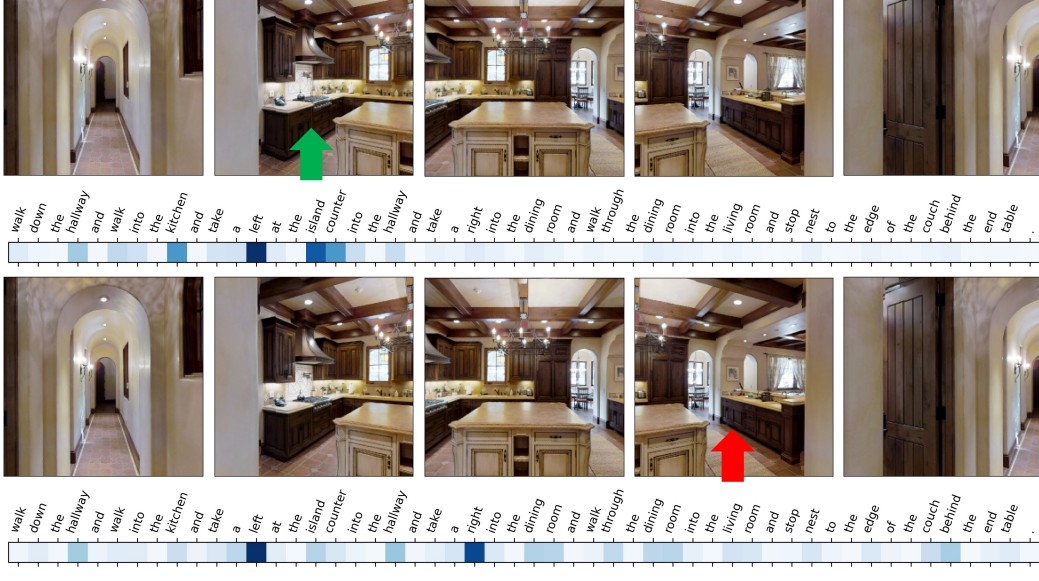

Figure 3: Navigation examples in normal scenes of the FDA-enhanced and baseline models.

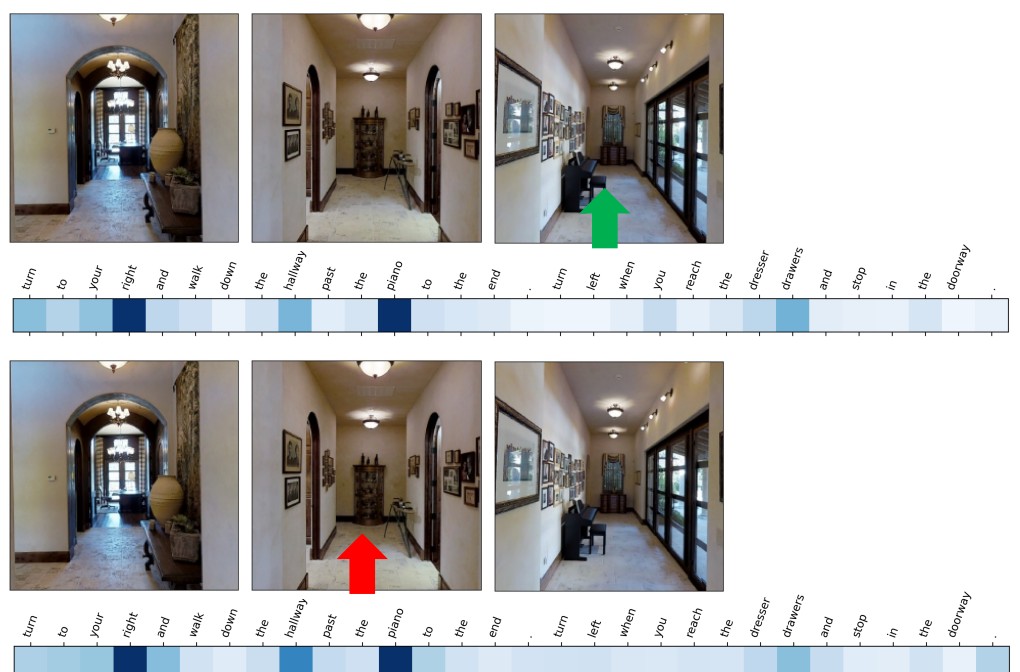

Figure 4: Navigation examples in normal scenes of the FDA-enhanced model and baseline models.

Figure 3 and Figure 4 show two navigation examples in normal scenes of the FDA-enhanced (the first row) and baseline (the second row) models. Figure 3 shows that the FDA-enhanced model could better find the essential words, namely "island counter" for visual-textual matching and ultimately aid the model in making correct action decisions. In the examples shown in Figure 4, both models obtained similar textual attention. However, unlike our model, the baseline model failed to choose the correct action. This indicates that the FDA-enhanced model, leveraging its enhanced capacity to capture high-frequency information, exhibits superior performance in cross-modal matching and consequently facilitates improved navigation decisions.

Figure 5, Figure 6 and Figure 7 depict three common navigation failure cases in FDA-enhanced model, each corresponding to the visual interference from similar objects, textual interference from

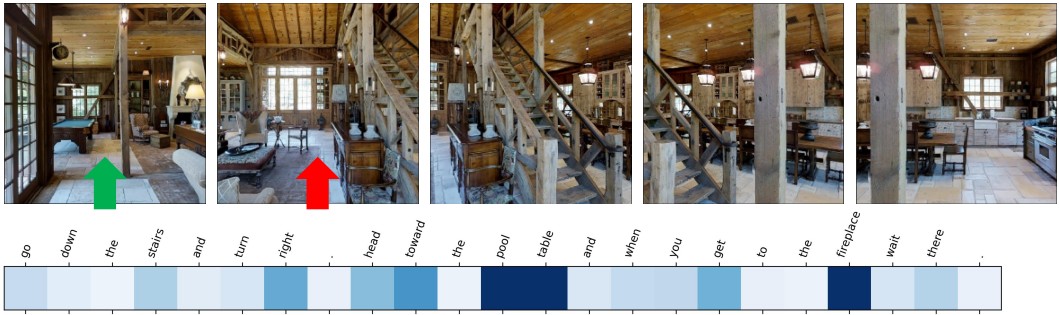

Figure 5: Navigation failure case: visual interference from similar objects.

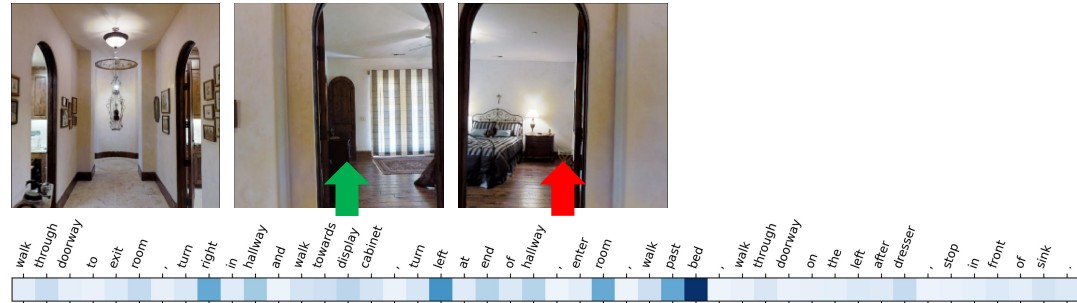

Figure 6: Navigation failure case: textual interference from adjacent object-related and direction-related words.

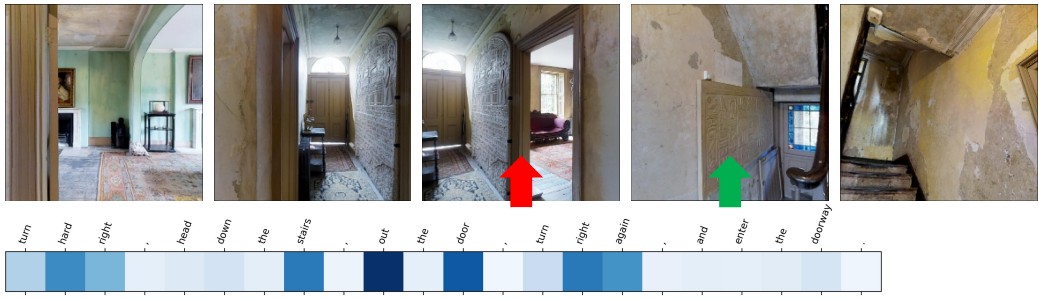

Figure 7: Navigation failure case: textual interference from adjacent object-related words.

adjacent object-related and direction-related words, and textual interference from adjacent object-related words. In Figure 5, the table in the second view bears a significant visual similarity to the pool table. Consequently, the agent is misdirected towards the wrong direction during cross-modal alignment. In Figure 6, according to the given instruction, the agent should turn left to enter the room corresponding to the second view. However, due to the influence of the adjacent word "bed" in the instruction, when the agent sees a bed in the wrong direction, it bypasses the left turn and proceeds directly to the room where the bed is located. Figure 7 illustrates the scenario where the agent prioritizes proceeding towards the object "door" that appears later in the instruction, instead of the earlier mentioned "stairs". This leads to a deviation in the navigation. These failure cases indicate that the next step for navigation improvement may lie in finer-grained discrimination for similar objects, as well as focusing on the temporal aspect of navigation reasoning through more precise text parsing.

## D   Implementation Details

For HAMT on R2R and RxR, we set the learning rate (lr) as 1e-5 and the batch size (bz) is 8. And the training iteration is 200K. In the case of HAMT on CVDN, the lr, bz and training iteration are set

as 1e-5, 8 and 100k, respectively. For DUET, we first pretrain the model as guided by [1] and then finetune it on R2R, with lr set as 1e-5 and bz set as 8. The total iteration for the finetuning is 50k. For TD-STP, the lr and bz are set as 1e-5 and 8. The training iteration is 200K. We set the optimizer as AdamW [4] for all models. The visual feature for R2R, CVDN and REVERIE are extracted by the ViT-B/16 [2], and the visual feature for RxR is extracted by the CLIP [5]. The textual encoder is aligned with the baseline models. In addition, the style-transfer-based ENVEDIT feature [3] we adopt use the fixed style embedding across 36 discretized views in a panoramic setting. All the experiments are conducted with a single NVIDIA RTX 3090 GPU. The validation and test phases are under the single-run setting.

# E   A Revisit to Discrete Fourier Transform on RGB Image

Considering an image $X \in \Re^{3 \times H \times W}$ including RGB color channels, $X^c$ denotes the image with the single color channel c, where $c \in \{r, g, b\}$. The corresponding frequency spectrum $F^c$ can be obtained by the Fourier Transform (FFT) formula $\mathcal{F}(I^c)$ on image $X^c$:

$$F_X^c[m,n] = \mathcal{F}(X^c)[m,n] = \sum_{h=0}^{H-1} \sum_{w=0}^{W-1} X^c[h,w] \cdot exp(-iuh) \cdot exp(-ivw), \ \ u = \frac{2\pi}{H}m \ \ v = \frac{2\pi}{W}n \quad (1)$$

where $F_X^c[m,n]$ denotes the value at the position $[m,n]$ of frequency spectrum $F_X^c$, $X^c[h,w]$ denotes the value at the position $[h,w]$ of image $X^c$, $u$ and $v$ are verticle frequency and horizon frequency, respectively. Similarly, the inverse Fourier Transform (iFFT) $\mathcal{F}^{-1}(F_X^c)$ maps the frequency spectrum $F_X^c$ to the image domain, resulting in the original image $X^c$:

$$X^c[h,w] = \mathcal{F}^{-1}(F_X^c)[h,w] = \frac{1}{HW} \sum_{m=0}^{H-1} \sum_{n=0}^{W-1} F_X^c[m,n] \cdot exp(iuh) \cdot exp(ivw), \ \ u = \frac{2\pi}{H}m \ \ v = \frac{2\pi}{W}n. \quad (2)$$

The frequency spectrum $F_X^{\{rgb\}}$ and $X$ can be obtained by stacking the frequency spectrums $\{F^r, F^g, F^b\}$ and images $\{X^r, X^g, X^b\}$ of RGB channels together, respectively:

$$F_X^{\{rgb\}} = Stack(F^r, F^g, F^b), \ \ X = Stack(X^r, X^g, X^b). \quad (3)$$