# OpenReview forum: "Frequency-Enhanced Data Augmentation for Vision-and-Language Navigation"
_NeurIPS.cc/2023/Conference — NeurIPS 2023 poster_

### Official Review · Reviewer_bEiK · 2023-07-03

**Soundness:** 3 good
**Presentation:** 3 good
**Contribution:** 3 good
**Rating:** 6
**Confidence:** 4

**Summary:**

Authors propose a Frequency enhanced Data Augmentation (FDA) technique to improve generalization/robustness to unseen environments in the vision-language navigation task. This work provides an initial analysis that shows high frequency information impacts performance. Results are provided on R2R, RxR, and CVDN datasets and are better than prior work when using the same models.

**Strengths:**

- The paper is clearly written and easy to follow.
- Experimentally the authors demonstrate that the proposed FDA data augmentation technique improves performance on most of the models and validation splits.

**Weaknesses:**

- The premise of the paper is slightly confusing. The high and low frequency perturbed images essentially are different domains, so of course performance drops. Additionally, counter to what the writing of the paper details, low frequency information also helps on SR for the val seen split in Table 1, and the unseen results vary but < 1 point on SR.
- FDA can hurt performance or doesn’t improve as much on seen compared to unseen results (e.g., R2R DUET, val seen).
- Table 4. It seems like an obvious result that a method trained with FDA is going to do better on high frequency perturbed images, this table may not be necessary or at least does not need to be included in the main text. Are there downstream scenarios where this kind of perturbation would even realistically happen?
- While many types of data augmentation are referenced in the related work section, the authors only experimentally compare to one (ENVEDIT) data augmentation work, and do not try baselines with other data augmentations that may affect similar information to the high frequency perturbation. Because of this, it feels the experiments are incomplete, and generally underwhelming due to the small SR performance changes.


**Questions:**

**Questions:**
- Figure 1 and the original analysis that founds this paper begs the question: how much of the performance effect shown by high frequency perturbation is caused by loss of color in the resulting image? What if you made the image black and white, or the low frequency image black and white? The analysis feels lacking to truly demonstrate that it is frequency information that matters.
- Did you try other methods of fusing the high frequency information into the training images? Or other training regimes instead of alternating the images per even and odd training steps per Eq. (6)?
- Are all of the results for a single experiment run?

**Comments and Suggestions:**
- L38-L40, I suggest notating where in the Figure to look. I.e. "Figure 1 (middle)" or "Figure 1 (right)" instead of repeating just "Figure 1".
- L59-64 is a very long single sentence, it should be broken up to several shorter sentences to make it easier to follow.
- Figure 4: there is room within the figure to have an abbreviation legend since there are a ton of acronyms in the figure that are not defined in the figure caption - i.e., what is GHPF? FFT? etc. They are defined in earlier figures but it would be better to have them defined within the scope of this figure, too.
- It would be better if figures appeared on pages they were referenced.
- L111-118, I am not really sure what the difference is between the points being made in 2) and 3).
- The notation in Eq. (2): It switched between using the $F$ notation for frequency and the $\mathcal{F}$ notation for FFT. Isn’t the high and low frequency $H$ on the original image $I$, while $\hat{H}$ should be on the interference image $\hat{I}$? Right now $H$ and $\hat{H}$ have the same definition in the equation.
- L156 "Lengt" --> "length", L153 "7k" --> "7,000", no comma in "2050" --> "2,050"
- L158 "Process" --> "Progress"?
- L178 says GP performance increases by 3.6, but the table shows a **0.36** improvement.
- L230 add a space after "et al."; generally throughout paper this is an issue.

**Limitations:**

The initial analysis prompting this work is not very convincing of the importance of high frequency information in images, and the paper does not compare to multiple data augmentation techniques. It could have additional baselines for simple image augmentations or other augmentation methods from prior work. This is particularly important because the performance improvement is not very large across the different models and datasets. Given the small performance differences, if the reported results are from a single experiment it is not convincing that the FDA approach is consistently effective.

---

> ### Author Rebuttal · Authors · 2023-08-10
>
> **Answer to Weakness-1:** Thanks for your comments. Figure 2 reveals important insights into the behavior of baseline models: HAMT, DUET, and TD-STP. These models show relatively high SR under low-frequency perturbations but experience significant SR drops under high-frequency perturbations. This indicates VLN models’ deficiency in effectively capturing the required high-frequency information.
>
> In Table 1, the small 0.19 SR improvement on val seen might stem from overfitting, given the significant 0.9 SR drop on val unseen. In VLN, navigation ability is primarily evaluated using unseen environments for better generalization. On val unseen, TD-STP+high frequency shows a 0.72 SR increase over the baseline and a 1.62 improvement compared to TD-STP+low frequency. This indicates high-frequency plays a more crucial role than low-frequency in cross-modal navigation.
>
> Considering the above-mentioned reasons, we design our approach to enhance the agent’s ability to capture crucial high-frequency information necessary for cross-modal navigation.
>
> Moreover, an improvement in SR and SPL on a strong (SoTA) baseline (SR 69.65, SPL 62.97), with gains of 0.72 and 0.65, reaching 70.37 and 63.62, constitutes a noteworthy advancement. Furthermore, TD-STP+HF outperforms TD-STP+LF by 1.62/1.66 in SR/SPL on val unseen, showcasing substantial enhancement.
>
> **Answer to Weakness-2:**  Thanks for your concern. The mentioned phenomenon results from severe overfitting of DUET to val seen. This overfitting issue in seen environments distorts the model’s true navigational capability assessment. Thus, seen environments have less significance as a reference in VLN. Instead, unseen environments are the main way to evaluate navigation ability, as they best reflect generalization capacity. On the VLN official competition leaderboards, agents are only evaluated and ranked in unseen environments.
>
> Hence, our optimal model choice is solely based on val unseen performance. In practice, during training, DUET+FDA exhibited 78.55/71.48 SR on val seen/unseen. Due to its underperformance on the val unseen compared to the paper’s reported model (76.00/71.86 on validation seen/unseen), its navigational ability is deemed inferior. For val unseen, our method, applied to HAMT, DUET, and TD-STP, notably enhances SR by 1.88/1.66/2.13, respectively.
>
> **Answer to Weakness-3:**  Table 4 is to illustrate FDA-enhanced models excelling in both the unseen perturbed images (val unseen) and the unseen perturbation images (ImageNet). While such scenarios may not exist in reality, they serve as compelling evidence of our method’s remarkable capability to capture essential high-frequency information for cross-modal navigation. This consideration led us to include it in the main manuscript.
>
> **Answer to Weakness-4-Part1:** Since no existing VLN data augmentation method aligns with our approach, which uniquely requires no additional parameters, data, or external models, comparisons with such methods are not available. Instead, we conduct a comparison with ENVEDIT, the only method sharing a similar image-level data augmentation strategy with ours.
>
> For reference, we’ve added comparisons between the FDA method with two other methods: Speaker [11*] and PREVALENT [13*], as shown in Table 1 of the attached PDF, using Baseline-1 [11*] and Baseline-2 [53*] models (* indicates the reference number in the main paper.). The result shows that our method achieves an additional gain of 5.2 compared to the Speaker based on extra model. Furthermore, our approach can effectively complement the two data augmentation methods.
>
> **Answer to Weakness-4-Part2-[small SR performance changes]:**  Our method applied to HAMT, DUET, and TDSTP significantly boosts val unseen performance of these SoTA methods, with SR increases of 1.88/1.66/2.13 respectively. Our improvements compared to the previous SoTA methods (DUET, TD-STP) rival or even surpass that of previous studies [29*, 37*, 50*, 1, 2, 3] (mainly below 2 points). Additionally, we experiment on two LSTM-based models: Baseline-1 [11*] and Baseline-3 [39*], as depicted in Table 2 of the attached PDF. These two models represent impressive 7.3/3.4 improvements.
>
> [1] KERM: ... CVPR. 2023.
>
> [2] GeoVLN: ... CVPR. 2023.
>
> [3] Local Slot Attention for VLN. CVPR. 2023.
>
> **Answer to Question-1:** We use the method in Figure 4 to generate high-frequency interference (HFI) and low-frequency interference (LFI) images for Figure 2. See the “Augmented Image" in Figure 4 for the HFI image. Since the image with HFI retains low-frequency components where the color information mainly resides (as illustrated in Figure 1), the color remains largely intact in the HFI image. However, in Figure 2, the model performs notably worse with HFI scenarios compared to LFI scenarios. This underscores that high-frequency, not color, is a key factor influencing cross-modal navigation.
>
> **Answer to Question-2:** We attempted to introduce high-frequency interference using weighted fusion on original images. The SR on val unseen was 70.54, notably below our method’s 71.78. Thus, we chose the current approach. Additionally, we experimented with random training on original and augmented images during navigation. The result is slightly inferior to the current method, with a 71.65 SR.
>
> **Answer to Question-3:** In Appendix Section B, we experiment with four seeds. Results consistently show notable navigation performance enhancement. SR improves from 1.74 to 2.89, with a mean increase of 2.13.
>
> **Answer to Comments and Suggestions:** Regarding 2) and 3) (L111-118): 2) highlights high-frequency information’s role in enhancing robustness for performance under environmental variations. 3) shows high-frequency information’s role in reducing overfitting for performance in new environments. We sincerely appreciate your thoughtful feedback regarding the issues with formulas, spelling, formatting, and expression. We will earnestly make the necessary revisions as suggested.

---

> > ### Comment · Reviewer_bEiK · 2023-08-14
> > **Rebuttal**
> >
> > I'd like to thank the authors for answering my questions and for the additional experiments they have included that make the paper results more compelling. After reading the rebuttal and answers to other reviewers' questions, I will be raising my score to weak accept.

---

> > > ### Author Response · Authors · 2023-08-15
> > >
> > > Dear Reviewer bEiK,
> > >
> > > We greatly appreciate your diligent review of this paper.  We are delighted to learn that our efforts could well address your concerns.  Thanks so much for your recognition!
> > >
> > > Best wishes,
> > >
> > > Paper12863 Authors

---

### Official Review · Reviewer_LWMZ · 2023-07-06

**Soundness:** 4 excellent
**Presentation:** 4 excellent
**Contribution:** 2 fair
**Rating:** 6
**Confidence:** 5

**Summary:**

In this paper, the authors propose a Frequency-enhanced Data Augmentation (FDA) for VLN models to enhance their ability to capture high-frequency information, thus improving their performance. First, at each step, they take the view as Reference Image and randomly sample an image as Interference Image. Then they transform these two images to the frequency domain space via FFT and mix the high-frequency components of two images. Finally, they obtain the augmented image via iFFT and alternately use original views’ images and augmented images to train the agent. the Experiments on different benchmarks reveal the proposed augmentation method is helpful to VLN models. However, I have some concerns about this paper. My detailed comments are as follows.

**Strengths:**

1. The authors take an analysis of frequency information domain by fuse images’ high-frequency or low-frequency components into original images, and find the importance of high-frequency for the agent’s better performance in VLN tasks.

2. The authors propose a simple data augmentation method, Frequency-enhanced Data Augmentation (FDA), which needs no external models and does not add complexity. The proposed method enhances the models’ ability to capture necessary high-frequency information and has been demonstrated to improve the models’ performance on multiple datasets.

**Weaknesses:**

1. I wonder why the authors choose to incorporate the high-frequency components from randomly sampled interference images into the original image, rather than using a fixed interference image, or utilizing the high-frequency components of the image itself, or randomizing certain high-frequency components of the original image. It would be better if the author could discuss these approaches or include them in the ablation experiments.

2. While the importance of high frequency compared to low frequency has been demonstrated by other experiments, it would be beneficial to conduct an ablation experiment utilizing low frequency for data augmentation under the same settings.

3. Some novel SoTA models are not shown in the tables in this paper, such as Airbert [1] on R2R benchmark. It would be better for the authors to add these state-of-the-art models into the comparative experiments which would demonstrate the strengths and limitations of the proposed method more comprehensively. Also, it is better to add the brief introduction about the VLN works on continued environment, such as [2,3] for the sake of completeness

4. There appear several low-level calculation/statistical errors that should have been avoided in the paper. For instance, on line 216, it should be "4 and 3 points," and on line 222, it should be "3.3 and 3.3 (2 and 1.7)." Additionally, on line 178, it should be "boost the GP performance by 0.36." The author should carefully review the data in the tables and the calculations presented throughout the text.


**Minor issues

1. In Equation 2, since Equation 1 already uses $F_{\hat{I}}^{\{rgb\}}$ to represent the operation result of $\mathcal{F}^{\{rgb\}}(\hat{I})$, it would be better to use $F_{\hat{I}}^{\{rgb\}}$ instead of $\mathcal{F}^{\{rgb\}}(\hat{I})$ in Equation 2.

2. On page 5, line 156, “Path Lengt” should be “Path Length”.

3. On page 6, line 178, “boost the GP performance by 3.6” should be “boost the GP performance by 0.36”.

4. On page 8, line 216, “by 4 and 2 points” should be “by 4 and 3 points”.

5. On page 8, line 222, “by 3.4 and 3.3” should be “by 3.3 and 3.3”.

[1] Airbert: In-domain Pretraining for Vision-and-Language Navigation, ICCV 2021

[2] Weakly-Supervised Multi-Granularity Map Learning for Vision-and-Language Navigation, NeurIPS 2022.

[3] Cross-modal Map Learning for Vision and Language Navigation, CVPR 2022.

**Questions:**

The manuscript is well organized and the proposed consistent improvement upon several strong baselines on 3 datasets. It would be better for the authors to evaluate the proposed FDA technique on other multi-model tasks, not limited to the VLN task.

**Limitations:**

The authors have adequately addressed the limitations

---

> ### Author Rebuttal · Authors · 2023-08-10
>
> **Q1: I wonder why the authors choose to incorporate the high-frequency components from randomly sampled interference images into the original image, rather than using a fixed interference image, or utilizing the high-frequency components of the image itself, or randomizing certain high-frequency components of the original image. It would be better if the author could discuss these approaches or include them in the ablation experiments.**
>
> A1: Thank you for your intriguing suggestion. The reason we randomly incorporate the high-frequency components of the interference image into the original image is to provide a more diverse set of high-frequency negative samples. In response to your suggestion, we conduct experiments under three settings: (1) Setting 1 - only using a fixed interference image’s high-frequency components as interference; (2) Setting 2 - using the high-frequency components of one channel in the image as interference for other channels; (3) Setting 3 -randomizing certain high-frequency components of the original image. In Setting 1, the model achieves SR score of 70.92. In Setting 2, the score is 69.26, and in Setting 3, the score is 69.73. Compared to the baseline model whose SR is 69.65, we observe a slight decline in performance in Setting 2, a slight improvement in Setting 3, and the most significant improvement in Setting 1. This suggests that incorporating high-frequency negative samples from other images effectively enhances the model’s ability to capture the required high-frequency information and improve cross-modal alignment. Comparing our method with Setting 1, we achieve superior performance with SR score of 71.78. The diverse sampling of high-frequency negative samples further improves the model’s ability to capture the necessary high-frequency information and enhance cross-modal alignment. We will add this discussion to the revision as suggested.
>
> **Q2: While the importance of high frequency compared to low frequency has been demonstrated by other experiments, it would be beneficial to conduct an ablation experiment utilizing low frequency for data augmentation under the same settings.**
>
> A2: Thanks for your valuable suggestion. In this experiment, we achieve an SR of 69.52. The result exhibits a minor decrement when compared to the baseline’s SR of 69.65, and notably trails behind the remarkable SR of 71.78 attained through high-frequency-based data augmentation. These findings will be incorporated into the revised version as recommended.
>
> **Q3: Some novel SoTA models are not shown in the tables in this paper, such as Airbert [1] on R2R benchmark. It would be better for the authors to add these state-of-the-art models into the comparative experiments which would demonstrate the strengths and limitations of the proposed method more comprehensively. Also, it is better to add the brief introduction about the VLN works on continued environment, such as [2,3] for the sake of completeness。**
>
> A3: Thanks for your valuable suggestion. We apologize for the oversight. Airbert [1] is an outstanding work, and we will include a comparison with it in the revised version. Additionally, as per your suggestion, we will introduce an overview of research on VLN-CE, such as [2,3].
>
> [1] Guhur, Pierre-Louis, et al. Airbert: In-domain pretraining for vision-and-language navigation. ICCV. 2021.
>
> [2] Chen, Peihao, et al. Weakly-supervised multi-granularity map learning for vision-and-language navigation. NeurIPS. 2022.
>
> [3] Georgakis, Georgios, et al. Cross-modal map learning for vision and language navigation. CVPR. 2022.
>
> **Q4: There appear several low-level calculation/statistical errors that should have been avoided in the paper. For instance, on line 216, it should be "4 and 3 points," and on line 222, it should be "3.3 and 3.3 (2 and 1.7)." Additionally, on line 178, it should be "boost the GP performance by 0.36." The author should carefully review the data in the tables and the calculations presented throughout the text.**
>
> A4: Thank you very much for your thorough review. We will definitely correct these errors in the revised version.
>
> **Q5: Minor issues.**
>
> A5: Thank you very much for your attention to detail. We will correct these errors in the revision.
>
> **Q6: The manuscript is well organized and the proposed consistent improvement upon several strong baselines on 3 datasets. It would be better for the authors to evaluate the proposed FDA technique on other multi-model tasks, not limited to the VLN task.**
>
> A6: Thank you for your sincere and enlightening feedback. In response to your suggestions, we conduct additional experiment on the REVERIE task using the SoTA model DUET. Based on our method, the model’s performance on SR, SPL, RGS, and RGSPL metrics shows significant improvement compared to the baseline model, with gains of 2.69/3.54/3.18/3.45, from 44.65/32.92/28.57/21.01 to 47.34/36.46/31.75/24.46. The comprehensive improvement of the agent’s navigation capabilities in the R2R, RxR, CVDN, and REVERIE tasks thoroughly validates the effectiveness of our method. In the next phase of our research, we will further extend the findings of this study to a broader range of cross-modal tasks.

---

> > ### Comment · Reviewer_LWMZ · 2023-08-20
> > **Thanks for the response**
> >
> > We thank for the authors' response. The response solves all my concerns and thus I will raise my rating from 5 to 6.

---

### Official Review · Reviewer_r2mS · 2023-07-07

**Soundness:** 3 good
**Presentation:** 2 fair
**Contribution:** 3 good
**Rating:** 6
**Confidence:** 5

**Summary:**

In this paper, the authors investigate the VLN task from a Fourier domain perspective, which aims to enhance the vision-language matching ability of agents. A Frequency-enhanced Data Augmentation (FDA) technique is proposed to improve the capability of capturing high-frequency information, which is seen as an important factor for training VLN agents. The experimental results show that baseline models can be further improved by adding the FDA technique.

**Strengths:**

- This work investigates the VLN task from an alternative perspective, i.e., the Fourier domain.

- A reasonable FDA method is proposed to enhance the baseline model to capture high-frequency information without the need for additional parameters.

- With the FDA technique, the performances of three transformer-based methods are further improved, confirming its effectiveness.

**Weaknesses:**

- The adopted HAMT, DUET, and TD-STP are all transformer-based agents. However, the LSTM-based agents with FDA also need to be verified.

- In Table 2, DUET+FDA exhibits a significant performance drop on the val-seen set compared to the original DUET, e.g., SPL from 73.71\% to 69.66\%.

- Lacking experimental comparison on the high-level instruction REVERIE dataset.

- As shown in Table 2 of the supplementary material, different random seeds have a significant impact on the model performance. For example, seed-3 and seed-4 differ by 1\% in SPL.

- The writing needs to be further polished, e.g., the highest accuracy numbers in all tables should be bolded, and the title of section 3.4 should be 'Comparison with the State-of-the-art Methods'.

- Missing numerous references such as [1-4].

[1] Adaptive Zone-Aware Hierarchical Planner for Vision-Language Navigation. In CVPR 2023.

[2] HOP+: History-enhanced and Order-aware Pre-training for Vision-and-Language Navigation. In TPAMI.

[3] Reinforced Structured State-Evolution for Vision-Language Navigation. In CVPR 2022.

[4] LANA: A Language-Capable Navigator for Instruction Following and Generation. In CVPR 2023.

**Questions:**

See Weaknesses.

---

> ### Author Rebuttal · Authors · 2023-08-10
>
> **Q1: The adopted HAMT, DUET, and TD-STP are all transformer-based agents. However, the LSTM-based agents with FDA also need to be verified.**
>
> A1: Thanks for your valuable suggestion. We conduct experiments on two LSTM-based models: Agent [1] and Follower [2]. The first baseline model, Agent, achieves an SR of 51.1 on R2R val unseen. After applying our FDA enhancement, the SR of the model increases to 54.5, a notable 3.4 improvement. The second baseline model, Follower, achieves an SR of 38.3 on R2R val unseen. After FDA enhancement, the SR of the model increases to 45.6, representing an impressive 7.3 improvement. These results demonstrate the effectiveness of our method not only for Transformer-based models but also for LSTM-based models. We will include these suggested experiments in the revised version.
>
> [1] S. Shen, et al. How much can CLIP benefit vision-and-language tasks? ICLR. 2022.
>
> [2] D. Fried, et al. Speaker-follower models for vision-and-language navigation. NeurIPS. 2018.
>
> **Q2: In Table 2, DUET+FDA exhibits a significant performance drop on the val-seen set compared to the original DUET, e.g., SPL from 73.71 to 69.66.**
>
> A2: Thank you for your comment. The phenomenon mentioned is a result of severe overfitting of DUET to the val seen split. The issue of overfitting in seen environments can impact the assessment of the model’s true navigational capability. Thus, seen environments have less significance as a reference in the VLN task. Instead, VLN unseen environments are the main way to evaluate navigation ability. This is because navigating unseen environments best reflects the model’s generalization ability. On the VLN competition leaderboards (both R2R and RxR), agents’ navigation capabilities are only evaluated and ranked in unseen environments.
>
> Hence, we base our best model selection solely on VLN’s performance in the val unseen split. In practice, during the training process, DUET+FDA exhibited an SR of 78.55 on val seen and 71.48 on val unseen. Nevertheless, due to its underperformance in the unseen environments compared to the model reported (val seen SR 76.00, val unseen SR 71.86) in our paper, its navigational ability is considered inferior. On validation unseen split, our method, applied to HAMT, DUET, and TD-STP, substantially improves the SR by 1.88, 1.66, and 2.13, respectively. Notably, our method significantly improves the performance of TD-STP to achieve a new SoTA performance.
>
> Moreover, our method can effectively mitigate overfitting in models. It reduces the gap in SR between the seen and unseen splits compared to the baseline models, reducing it from 9.38, 8.35, and 6.65 to 8.29, 4.14, and 5.89, respectively.
>
> **Q3: Lacking experimental comparison on the high-level instruction REVERIE dataset.**
>
> A3: Thanks for your valuable suggestion and we will add it to our revision. For your reference, we have conducted experiments on the REVERIE dataset using the SoTA model DUET. Based on our method, the model’s performance on SR, SPL, RGS, and RGSPL metrics shows significant improvement compared to the baseline model, with an increase of 2.69/3.54/3.18/3.45, from 44.65/32.92/28.57/21.01 to 47.34/36.46/31.75/24.46. The comprehensive improvement of the agent’s navigation capabilities in the R2R, RxR, CVDN, and REVERIE tasks thoroughly validates the effectiveness of our method.
>
> **Q4: As shown in Table 2 of the supplementary material, different random seeds have a significant impact on the model performance. For example, seed-3 and seed-4 differ by 1 in SPL.**
>
> A4: Thank you for your comment. The fluctuations observed in the navigation metrics within Table 2 of the Appendix are considered normal. The average values for SR and SPL are 71.78 and 63.64, respectively. The maximum deviations of SR and SPL from their respective means, across the four experimental groups, are only 0.76, which is a reasonable value. We notice that HAMT [1] and EnvAg [2] also reported multiple experimental results, and our deviation values are quite similar to them. SR deviation reaches 1.4 and 2.14 for HAMT and EnvAg, while SPL deviation hits 1.1 and 2.12 for the same.
>
> [1] S. Chen, et al. History aware multimodal transformer for vision-andlanguage navigation. NeurIPS. 2021.
>
> [2] X. E. Wang, et al. Environment-agnostic multitask learning for natural language grounded navigation. ECCV. 2020.
>
> **Q5: The writing needs to be further polished, e.g., the highest accuracy numbers in all tables should be bolded, and the title of section 3.4 should be ’Comparison with the State-of-the-art Methods’.**
>
> A5: Thank you for your valuable suggestions. We are committed to earnestly incorporating your suggestions to enhance the clarity and precision of our expression.
>
> **Q6: Missing numerous references such as [1-4].**
>
> A6: Thank you for the reminder. We sincerely apologize for overlooking these four outstanding and interesting works. Among them, two CVPR 2023 papers were not published at the time of our submission. We will include proper citations for all of them in the revised version.
>
> [1] Gao, Chen, et al. Adaptive Zone-Aware Hierarchical Planner for Vision-Language Navigation. CVPR. 2023.
>
> [2] Qiao, Yanyuan, et al. HOP+: History-enhanced and Order-aware Pre-training for Vision-and-Language Navigation. TPAMI. 2023.
>
> [3] Chen, Jinyu, et al. Reinforced structured state-evolution for vision-language navigation. CVPR. 2022.
>
> [4] Wang, Xiaohan, et al. LANA: A Language-Capable Navigator for Instruction Following and Generation. CVPR. 2023.

---

> > ### Comment · Reviewer_r2mS · 2023-08-20
> > **Post rebuttal**
> >
> > The authors solve all my concerns in the response, and I believe this work has good insights for the community. Thus I decide to raise my score to 6.

---

### Official Review · Reviewer_gjxf · 2023-07-09

**Soundness:** 2 fair
**Presentation:** 4 excellent
**Contribution:** 3 good
**Rating:** 6
**Confidence:** 5

**Summary:**

This paper aims at investigating the Fourier domain of vision-and-language navigation (VLN). They propose Frequency-enhanced Data Augmentation (FDA) to improve the visual-text matching processing based on high-frequency visual information. Their method can achieve promising results efficiently on various VLN datasets, including R2R, RxR, and CVDN, with the same parameter size.

**Strengths:**

+ This paper is well-written and easy to follow.
+ The aspect of the Fourier domain and lower-/higer- frequency visual representation is new in VLN.
+ Though data augmentation is not novel in VLN, the frequency-based sampling seems effective, which helps achieve new/competitive SOTA on various datasets.
+ The sensitive analysis in Figure 2 and Table 1 is a good intro convincing.

**Weaknesses:**

+ Though the performance looks effective, what is the motivation to rely on Fourier representation instead of the widely-used learned visual features (maybe I missed it, but I cannot find a clear answer in the draft)?
+ Does visual information get lost during the Fourier transform? How to overcome this issue during the navigation process?
+ A detailed analysis of data augmentation should be considered.  For example, how much data can lead to how much improvement? If we keep training with more data, does the navigator works better and better?
+ There should include some qualitative examples to demonstrate how Fourier features can actually lead to the final goal. The comparison with baselines is required, and both successful/failed cases should be involved to guide future research.

**Questions:**

Please see Weakness

---

> ### Author Rebuttal · Authors · 2023-08-10
>
> **Q1: Though the performance looks effective, what is the motivation to rely on Fourier representation instead of the widely-used learned visual features (maybe I missed it, but I cannot find a clear answer in the draft)?**
>
> A1: Thank you for your question. Our method is essentially a combination of Fourier transform and learned visual features. In data augmentation phase, we rely on Fourier transform to obtain the high and low-frequency information of the reference image and the interference image. By using the Mix operation (Formula 3) and Fourier inverse transforms, we generate the augmented image. However, during the training and testing phases, we do not use Fourier transforms. Instead, we utilize learned visual features. In the training phase, we feed the reference image and augmented image into the image encoder to extract learned visual features. Subsequently, we input these features into the baseline model to learn downstream visual features and perform navigation reasoning based on them. During the test phase, we directly extract visual features from the reference image using the image encoder and feed them into the FDA-enhanced model to accomplish navigation reasoning. Because of this characteristic of our method, our lightweight approach can easily enhance existing navigation models based on learned visual features, such as TD-STP, DUET and HAMT. We apologize for any unclear expressions and will ensure clarity in the revised version.
>
> **Q2: Does visual information get lost during the Fourier transform? How to overcome this issue during the navigation process?**
>
> A2: Thanks for pointing out your concern. We implement the Fourier forward and inverse transforms using numpy.fft.fftn and numpy.fft.ifftn, which almost incur no information loss. Meanwhile, our method enhances the ability to capture high-frequency information of the model itself. As a result, during the test phase, the model directly processes RGB images, same to its baseline model, without relying on Fourier transform.
>
> **Q3: A detailed analysis of data augmentation should be considered. For example, how much data can lead to how much improvement? If we keep training with more data, does the navigator works better and better?**
>
> A3: Thanks for your constructive suggestion. In order to identify some underlying patterns, we conduct a preliminary exploratory experiment using current augmented data volume’s 10%, 100%, and 200%. Compared to the baseline model, which achieves an SR of 69.65, the model trained with 10% augmented data exhibits an SR of 69.77. Similarly, the model trained with 100% augmented data achieves an SR of 71.78, and the model trained with 200% augmented data achieves an SR of 71.86. As the volume of augmented data increases, the model’s performance gradually improves; however, this improvement trend begins to plateau. We will add this discussion to the revision.
>
> **Q4: There should include some qualitative examples to demonstrate how Fourier features can actually lead to the final goal. The comparison with baselines is required, and both successful/failed cases should be involved to guide future research.**
>
> A4: Thanks for your valuable suggestions. Due to the fact that our method does not require the Fourier features during navigation, but only relies on learned visual features, it is not possible to perform a direct qualitative analysis of Fourier features. However, we have still attempted to analyze the results from other perspectives. Firstly, we conduct navigation experiments under high-frequency interference settings (Section 3.3) and provide corresponding visualizations (Appendix - Figure 1, Figure 2). The results confirm that our model can effectively capture the necessary high-frequency information from images. Secondly, we provide visualizations of visual and textual matching (Appendix - Figure 3, Figure 4), which demonstrates that models with stronger high-frequency capturing abilities can utilize the captured high-frequency information to improve cross-modal attention weight assignment and cross-modal matching. Following your suggestions, we will add failed cases in the revised version and make every effort to provide in-depth analysis to assist other researchers in the community with their future research endeavors.

---

### Official Review · Reviewer_wftp · 2023-07-09

**Soundness:** 3 good
**Presentation:** 4 excellent
**Contribution:** 3 good
**Rating:** 6
**Confidence:** 4

**Summary:**

This paper investigates the sensitivity of existing state of the art VLN models to high frequency perturbations in the image features, and identifies that this could be a way to improve the models performance by making them more robust. It also provides evidence to show that adding high frequency information explicitly to the models may improve their performance. Taken together, this implies that if the model can be made more sensitive to changes in high frequency components while being able to discriminate which ones are relevant to the instruction, it could improve the generalization power of the models. The paper then proposes a data augmentation approach which involves replacing high frequency components of images with those of randomly selected other images, and training the model on this data. The resulting models degrade less in performance when evaluated on perturbed images and improve performance on standard benchmarks compared to the baseline models.

-------
Edit: Updated score after clarifications from the rebuttal!

**Strengths:**

## Originality
As far as I know, this is a new exploration in terms of data augmentation by perturbing visual features in the frequency domain

## Quality, Clarity, Significance
The paper provides an interesting exploration and provides evidence that supports their hypothesis. It is well written and easy to follow. The paper provides experiments based on three state of the art models, and provides a lightweight method to improve models' performance.

**Weaknesses:**

## Clarity
* Lacking in implementation details regarding choice of model components such as image encoder, text encoder, etc.

## Discrepancies in numbers reported
* The numbers reported in the table for RxR seem to not be consistent from those in literature (for example in terms of NDTW, HAMT on RxR Val seen achieves 65.3 and 63.1 on val unseen according to their paper [1] as opposed to the 63.1 and 59.94 reported here.
* In Table 5 (Comparison to EnvEdit), the numbers reported for HAMT + EnvEdit  for SR and SPL are 67.1 and 62.84 as opposed to 68.9 and 64.4 in the EnvEdit paper [2]
* Line 177: Text states that the improvement in GP is by 3.6 points whereas the table mentions 0.36 points.

Could the authors please explain these discrepancies?

## Minor issues / typos
* Line 156: Path Lengt -> Path Length
* Line 158: Goal Process -> Goal Progress
* Table 4 caption. Please state more clearly that this evaluation set is non-standard R2R, and is created as part of this paper


[1] History Aware Multimodal Transformer for Vision-and-Language Navigation. Shizhe Chen et al.
[2] ENVEDIT: Environment Editing for Vision-and-Language Navigation. Jialu Li et al.

**Questions:**

* What happens if you just do an interpolation between one of the channels from a random image, instead of doing it in the fourier domain? This could be additional proof that augmentation by manipulating in the fourier domain is most effective.
* According to Eq 3, the resulting image is likely to have perturbed features for more most of the channels - how did you decide the ratio for keeping original vs replacing with the distractor image features. Is this applied for all the views and all the images in the trajectory?
* What happens if you train only with the augmented data? what was the reason you chose the set up in (6)
* Since the edge and corner features are what this paper finds to be specifically useful, and given that the image feature extractors based on CNNs also extract such features in early layers, have the authors considered or tried reusing intermediate features from the image extractor to emphasize these features?

**Limitations:**

Yes

---

> ### Author Rebuttal · Authors · 2023-08-10
>
> **Q1: Lacking in implementation details regarding choice of model components such as image encoder, text encoder, etc.**
>
> A1: Thank you for your concern. Our model components closely match baseline models (TD-STP, DUET, HAMT). For R2R, TD-STP, DUET, and HAMT use ViT-B/16 for image encoder and BERT for text encoder. For CVDN, HAMT uses ViT-B/16 for image encoder and BERT for text encoder. For RxR, HAMT uses CLIP for image encoder and XLM-R for text encoder. As our focus is image-level data augmentation, we’ve detailed image encoders in Supplementary Section D. We’ll add text encoder information for better reader understanding.
>
> **Q2: The numbers reported in the table for RxR seem to not be consistent from those in literature (for example in terms of NDTW, HAMT on RxR Val seen achieves 65.3 and 63.1 on val unseen according to their paper [1] as opposed to the 63.1 and 59.94 reported here.**
>
> A2: Thanks for pointing this out. We anticipate that the reviewer erroneously refers to the incorrect numbers. In essence, our research aligns with the results in paper [1]. The numbers 65.3 and 63.1 in Table 13 of [1] are for val seen and val unseen, respectively. In our paper, the values 63.1 and 59.94 in Table 7 correspond to val unseen and test unseen, respectively. For [1]’s test unseen (nDTW=59.94), we suggest looking at its Table 12.
>
> [1] S. Chen, et al. History aware multimodal transformer for vision-andlanguage navigation. NeurIPS. 2021.
>
> **Q3: In Table 5 (Comparison to EnvEdit), the numbers reported for HAMT + EnvEdit for SR and SPL are 67.1 and 62.84 as opposed to 68.9 and 64.4 in the EnvEdit paper [2].**
>
> A3: Thanks for your question. We apologize for the unclear expression. ENVEDIT [2] implements a total of three types of data augmentation: Editing Style, Editing Appearance, and Editing Objects. The latter two require the semantic segmentation model to provide object and semantic information in addition to the generative model. The performance of 68.9 and 64.4 reported in [2] actually results from the ensemble of the three models trained on the three types of augmented ENVEDIT data, respectively. Unlike ENVEDIT, our method operates without external models. To ensure fairness, we compare our approach with Editing Style, which uses the fewest external models. [2] reports SR and SPL as 67.3 and 62.6, respectively. Our reimplementation confirms these as 67.31 and 62.84, seen in our Table 5. Our current statement (L205: “a strong data augmentation method ENVEDIT in VLN, which aims to increase the diversity of the environment using GANs to alter the environment style.") is not entirely clear, and we will enhance its clarity in our revisions.
>
> [2] J. Li, et al. Envedit: Environment editing for vision-and-language navigation. CVPR. 2022.
>
> **Q4: Line 177: Text states that the improvement in GP is by 3.6 points whereas the table mentions 0.36 points.**
>
> A4: We sincerely apologize for the oversight and greatly appreciate your attention to detail. The accurate value is indeed 0.36. We will rectify this error in the revised version.
>
> **Q5: Minor issues / typos.**
>
> A5: Thank you for your correction. We will further correct typos, and provide clearer clarifications in the revised version as suggested.
>
> **Q6: What happens if you just do an interpolation between one of the channels from a random image, instead of doing it in the fourier domain? This could be additional proof that augmentation by manipulating in the fourier domain is most effective.**
>
> A6: Thanks for your suggestion. The interpolation method achieves SR/SPL of 70.16/62.79 on R2R val unseen, notably lower than FDA’s 71.78/64.00. This indicates Fourier domain manipulation is superior since it can fully leverage high frequency for augmentation, which the spatial domain cannot achieve.
>
> **Q7: According to Eq 3, the resulting image is likely to have perturbed features for more most of the channels - how did you decide the ratio for keeping original vs replacing with the distractor image features. Is this applied for all the views and all the images in the trajectory?**
>
> A7: Thanks for your question. We empirically selected the current ratio, which has shown quite remarkable results. This ratio is consistently used for all views and images in the trajectory.
>
> **Q8: What happens if you train only with the augmented data?**
>
> A8: Thank you for bringing up this concern. The model trained only on augmented data achieves SR 67.01 and SPL 57.86 on R2R val unseen, performing worse than the model trained with both original and augmented data (SR 71.78, SPL 64.00). Exclusive reliance on augmented data may lack proper visual cues for the model to emphasize crucial high-frequency components, thus affecting cross-modal alignment.
>
> **Q9: What was the reason you chose the set up in (6)?**
>
> A9: Empirically, using the approach in setting (6) yields better performance (SR 71.78) than random selection of original and augmented data (SR 71.65). Thus, we opted for setting (6).
>
> **Q10: Since the edge and corner features are what this paper finds to be specifically useful, and given that the image feature extractors based on CNNs also extract such features in early layers, have the authors considered or tried reusing intermediate features from the image extractor to emphasize these features?**
>
> A10: Thanks for your interesting question. We conduct a preliminary experiment following a structure similar to Figure 3, using the layer1 features of CLIP-ResNet50. It showed an improvement over the baseline (SR 69.65), reaching SR 70.11. This underscores high-frequency information’s significance in cross-modal navigation. In contrast, our method achieves far better performance (SR 71.78) without additional parameters or model reliance.

---

### Author Rebuttal · Authors · 2023-08-10

Dear Chairs and Reviewers,

We would like to express our heartfelt gratitude to you for managing this paper and dedicating your valuable time to providing insightful comments. We also extend our sincere appreciation to the reviewers for recognizing the significance of our work. We have taken meticulous care in addressing all the concerns raised by the reviewers. For detailed information regarding the specific concerns, we kindly direct you to the Rebuttal Section.

The attached PDF file includes Tables 1-3 showing our experimental results:

1. Table 1 shows the comparison between our FDA method with two classic data augmentation methods: Speaker and PREVALENT.

    The result shows that our method, **relying on no additional parameters, data and external model**, achieves an **additional gain of 5.2 on SR** compared to the Speaker based on extra model. Furthermore, our approach can effectively **complement the two data augmentation methods**.

2. Table 2 represents the result of different LSTM-based models enhanced by the FDA method on R2R.


    The two LSTM-based models represent **impressive 7.3/3.4 improvements on SR**, demonstrating **the effectiveness of our method not only for Transformer-based models but also for LSTM-based models**.

3. Table 3 shows the results of our FDA method on the REVERIE task with the SoTA baseline DUET.

    Based on our method, the model’s performance on SR, SPL, RGS, and RGSPL metrics shows significant improvement compared to the baseline model, with **significant gains of 2.69/3.54/3.18/3.45**, from 44.65/32.92/28.57/21.01 to 47.34/36.46/31.75/24.46. The comprehensive improvement of the agent’s navigation capabilities in **the R2R, RxR, CVDN, and REVERIE tasks thoroughly validates the effectiveness of our method**.

If you have any further questions, we are more than willing to engage in further discussions.

Best wishes,

Paper12863 Authors

---

### Decision · Program_Chairs · 2023-09-21

**Decision:**

Accept (poster)

**Comment:**

This paper proposes Frequency-enhanced Data Augmentation (FDA) technique aimed to improve generalization, robustness, and performance on in VLN. Results on R2R, RxR, and CVDN empirically show that capturing critical high-frequency information w/ FDA helps.

The committee found the paper to be (1) well-written, (2) backed with strong suite of experiments across multiple base models and multiple VLN datasets, (3)  is a lightweight edit to improve performance across multiple base models, (3) investigates a novel aspect of this vision-language-embodiment task probing importance of high-frequency visual information and correspondence in text.

The discussion phase helped resolve concerns of the reviewers and revealed helpful updates to the draft.

The AC concurs with the unanimous decision of the reviewers and recommends acceptance. Also, the AC requests the authors to follow through on the promised revisions in the rebuttals (clarity, failure cases, new experiments, etc.).